# Leading edge maintenance in migrating cells is an emergent property of branched actin network growth

Rikki M Garner[1,2†], Julie A Theriot[2]*

[1]Biophysics Program, Stanford University School of Medicine, Stanford, United States; [2]Department of Biology, Howard Hughes Medical Institute, University of Washington, Seattle, United States

**Abstract** Animal cell migration is predominantly driven by the coordinated, yet stochastic, polymerization of thousands of nanometer-scale actin filaments across micron-scale cell leading edges. It remains unclear how such inherently noisy processes generate robust cellular behavior. We employed high-speed imaging of migrating neutrophil-like HL-60 cells to explore the fine-scale shape fluctuations that emerge and relax throughout the process of leading edge maintenance. We then developed a minimal stochastic model of the leading edge that reproduces this stable relaxation behavior. Remarkably, we find lamellipodial stability *naturally emerges* from the interplay between branched actin network growth and leading edge shape – with no additional feedback required – based on a synergy between membrane-proximal branching and lateral spreading of filaments. These results thus demonstrate a novel biological noise-suppression mechanism based entirely on system geometry. Furthermore, our model suggests that the Arp2/3-mediated ~70–80° branching angle optimally smooths lamellipodial shape, addressing its long-mysterious conservation from protists to mammals.

*For correspondence:
jtheriot@uw.edu

Present address: [†]Department of Systems Biology, Harvard Medical School, Boston, United States

Competing interest: The authors declare that no competing interests exist.

## Editor's evaluation

This paper describes analysis and modeling of leading edge fluctuations in migrating cells driven by a branched Arp2/3 lamellipodial network. A stochastic model shows how branching contributes to shape stability, and reproduces the measured spectrum and dynamics of leading edge fluctuations. Analysis of the model as a function of branching angle suggests that the Arp2/3 branching angle might be selected to smooth lamellipodial shape. This work provides new ideas to a big field of research, including Fourier analysis of leading edge fluctuations.

## Introduction

Cell migration driven by actin polymerization plays an essential role in countless organisms spanning the eukaryotic tree of life (*Pollard and Cooper, 2009*; *Fritz-Laylin et al., 2017a*; *Welch et al., 1997*). Across this broad phylogeny, cells have been observed to form a dizzying array of protrusive actin structures, each exhibiting unique physical and biological properties (*Svitkina, 2018*). In all cases, the fundamental molecular unit of these micron-scale structures is the single actin filament, which polymerizes stochastically by addition of single monomers to push the leading edge membrane forward (*Mogilner and Oster, 1996*; *Peskin et al., 1993*; *Theriot et al., 1992*; *Prass et al., 2006*). Higher order actin structures, and the biological functions they robustly enable, are therefore mediated by the collective action of thousands of stochastically growing filaments (*Svitkina, 2013*). It remains an open question how cells control for – or leverage – this inherent stochasticity to maintain stable

**eLife digest** In every human cell, there are tens of millions of proteins which work together to control everything from the cell's shape to its behavior. One of the most abundant proteins is actin, which organizes itself into filaments that mechanically support the cell and help it to move.

These filaments are very dynamic, with individual actin molecules constantly being added or removed. This allows the cell to build large structures with distinct shapes and properties. Many motile cells, for example, have a structure called a lamellipodium which protrudes at their 'leading edge' and pushes them forward. The lamellipodium has a very robust shape that does not vary much between different cell types, or change significantly as cells migrate. But how the tens of thousands of actin molecules inside the lamellipodium organize themselves into this large, stable structure is not fully understood.

To investigate, Garner and Theriot used high-speed video microscopy to track the shape of human cells cultured in the laboratory. As the cells crawled along a glass surface, their leading edge undulated like strings being plucked on a guitar. A computer simulation showed that these ripples can be caused by filaments randomly adding and removing actin molecules.

While these random movements could destabilize the structure of the leading edge, the simulation suggests that another aspect of actin filament growth smooths out any fluctuations in the lamellipodium's shape. Actin networks in the lamellipodium have a branched configuration, with new strands emerging off each other at an angle like branches in a tree. Garner and Theriot found that the specific angle in which new filaments are added smooths out the lamellipodium's shape, which may explain why this geometry has persisted throughout evolution.

These findings suggest that the way in which actin filaments join together helps to maintain the shape of large cellular structures. In the future, scientists could use this design principle to build molecular machines that can self-organize into microstructures. These engineered constructs could be used to modulate the activity of living cells that have been damaged by disease.

leading edge protrusions over length and time scales more than three orders of magnitude larger than the scales of actin monomer addition (*Rafelski and Theriot, 2004*; *Vavylonis et al., 2005*).

Perhaps the archetype of dynamically stable actin structures is the lamellipodium, a flat 'leaf-like' protrusion that is ~200 nm tall, up to 100 μm wide, and filled with a dense network of dendritically branched actin filaments (*Svitkina et al., 1997*; *Abraham et al., 1999*; *Laurent et al., 2005*; *Fritz-Laylin et al., 2017b*). Cell types that undergo lamellipodial migration (most notably fish epidermal keratocytes and vertebrate neutrophils) can maintain a single, stable lamellipodium for minutes to hours, allowing the cells to carry out their biological functions (*Tsai et al., 2019*; *Lacayo et al., 2007*). For example, in their in vivo role as first responders of the innate immune system, neutrophils must undergo persistent migration over millimeter-scale distances to reach sites of inflammation and infection (*de Oliveira et al., 2016*; *Kolaczkowska and Kubes, 2013*). Regardless of the cell type, the origins of this striking stability in the face of stochastic actin filament polymerization remain elusive. It has been widely been assumed for decades that some sort of regulatory or mechanical feedback mechanism must be required for lamellipodial shape stability, with extensive experimental efforts identifying membrane tension (*Diz-Muñoz et al., 2016*; *Houk et al., 2012*; *Mueller et al., 2017*; *Gauthier et al., 2012*; *Tsujita et al., 2015*; *Batchelder et al., 2011*; *Sens and Plastino, 2015*), plasma membrane curvature-sensing proteins (*Tsujita et al., 2015*; *Pipathsouk et al., 2019*), a competition for membrane-associated free monomers (*Mullins et al., 2018*), and force-feedback via directional filament branching (*Risca et al., 2012*) as potential contributors. The stability of lamellipodia has also been theoretically proposed to depend on the dendritically branched structure of their actin networks, wherein filaments are oriented at an angle relative to the cell's direction of migration, allowing growing filament tips to spread out laterally along the leading edge as they polymerize (*Lacayo et al., 2007*; *Grimm et al., 2003*). Although any acute angle would permit spreading, we note that filament orientation in cells has been experimentally observed to be highly stereotyped, averaging ±35° relative to the membrane normal (*Maly and Borisy, 2001*; *Verkhovsky et al., 2003*) – approximately one half of the highly evolutionarily-conserved ~70° branch angle mediated by the Arp2/3 complex (*Mullins et al., 1998*; *Volkmann et al., 2001*; *Rouiller et al., 2008*.) In contrast to

the proposed stabilizing role of spreading, several other features of lamellipodial actin are known to impart nonlinearities on network growth, which might amplify stochastic fluctuations. For instance, dendritic branching is an autocatalytic process which can lead to explosive growth (*Mullins et al., 2018*; *Carlsson, 2001*). In addition, the growth rates of the actin network are dependent on the velocity of the flexible membrane surface it is pushing, in a manner which imparts hysteresis to the system (*Mueller et al., 2017*; *Parekh et al., 2005*). How spreading might interact with these complexities – and the ultimate consequences for maintenance of a stable leading edge – remains unknown.

Seeking to dissect the origins of lamellipodial stability, we pursued complementary experimental and computational methodologies. First, we performed high-speed, high-resolution microscopy on migrating human neutrophil-like HL-60 cells to monitor their leading edge shape dynamics. In contrast to the remarkable overall lamellipodial stability observed over minutes, high-speed imaging revealed that the leading edge shape is extremely dynamic at shorter time and length scales, constantly undergoing fine-scale fluctuations around the average cell shape. We determined that these shape fluctuations continually dissipate (thereby enabling long time scale lamellipodial maintenance) in a manner quantitatively consistent with viscous relaxation back to the time-averaged leading edge shape. We next developed a minimal stochastic model of branched actin network growth against a flexible membrane, broadly applicable to a wide variety of cell types, that was able to recapitulate the global leading edge stability and fine-scale fluctuation relaxation behavior observed in cells. Our model suggests that the suppression of stochastic fluctuations is an intrinsic, emergent property of collective actin dynamics at the leading edge, as branched network geometry *alone* is necessary and sufficient to generate lamellipodial stability. Moreover, we find that the evolutionarily-conserved geometry, the ~70° branching angle of the Arp2/3 complex, optimally quells shape fluctuations.

## Results

### Fine-scale leading edge shape fluctuations revealed at high spatiotemporal resolution

Neutrophils form lamellipodia that are intrinsically lamellar, maintaining a thin, locally flat sheet of actin even in the absence of support structures like the substrate surface (*Fritz-Laylin et al., 2017b*). Here, we study the migration of neutrophil-like HL-60 cells (*Spellberg et al., 2005*) within quasi-two-dimensional confinement between a glass coverslip and an agarose pad overlay (*Millius and Weiner, 2009*). In addition to serving as an excellent in vitro model for neutrophil surveillance of tissues, this assay allows for easy visualization and quantification of lamellipodial dynamics by restraining the lamellipodium to a single imaging plane. Cells in this type of confinement can migrate persistently, maintaining nearly-constant cell shape, for time scales on the order of minutes to hours (*Tsai et al., 2019*; *Garner et al., 2020*). In order to capture leading edge dynamics on time scales more relevant to the stochastic growth of individual filaments, we performed high-speed (20 Hz) imaging of migrating HL-60 cells. These experiments revealed dynamic, fine-scale fluctuations around the average leading edge shape (*Figure 1a–c*, *Video 1*, *Figure 1—figure supplements 1–2*, Materials and methods), where local instabilities in the leading edge emerge, grow, and then relax. Notably, these previously-unobserved lamellipodial dynamics are phenotypically distinct from – and almost 100-fold faster than – the oscillatory protrusion-retraction cycles seen in other, slower-moving cell types (e.g. fibroblasts) (*Giannone et al., 2004*; *Ryan et al., 2012*; *Ma et al., 2018*).

We estimate we were able to reliably measure fluctuations with wavelengths as small as ~650 nm, and amplitudes down to ~65 nm, by fitting the phase contrast halo around the leading edge (*Figure 1—figure supplements 1–2*, Materials and methods). These values should approximately correspond to 25 actin filaments at physiological spacing (*Svitkina et al., 1997*) and 25 actin monomers assembled into a filament lattice along the direction of motion. While our measurements of shape dynamics cannot resolve polymerization events of individual filaments, our results are consistent with the hypothesis that stochasticity in actin growth at the level of monomer addition – occurring throughout the leading edge actin network – ultimately manifests as the observed micron-scale leading edge fluctuations. In particular, kymograph analysis of curvature and velocity (*Figure 1b–c*) showed that relatively long-lived shape fluctuations are formed by the continual time-integration of seemingly uncorrelated and very short-lived (sub-second, sub-micron) velocity fluctuations. Because the average cell shape remains constant over time, there must be some form of feedback acting

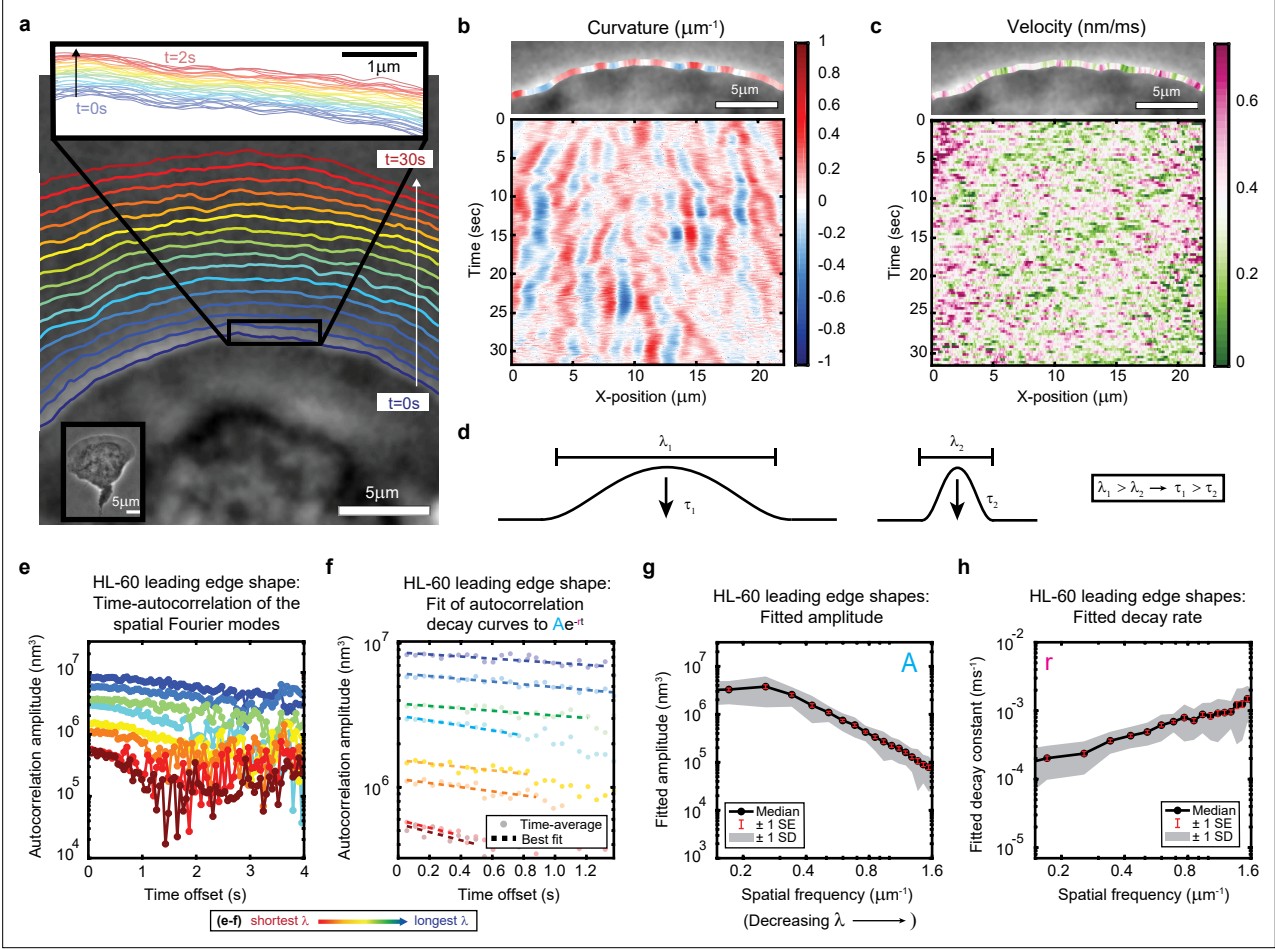

**Figure 1.** High-speed, high-resolution imaging reveals fine-scale fluctuations in leading edge shape. (**a–c**) Example of leading edge fluctuations extracted from a representative migrating HL-60 cell. (**a**) Phase contrast microscopy image from the first frame of a movie, overlaid with segmented leading edge shapes from time points increasing from blue to red in 2 s intervals. Top Inset: Magnification of the segmented leading edge between t = 0–2 s increasing from blue to red in 50 ms intervals. Bottom Inset: A de-magnified image of the whole cell at the last time point. (**b–c**) Kymographs of curvature (**b**) and velocity (**c**). Note the velocity is always positive, so no part of the leading edge undergoes retraction. (**d**) Schematic demonstrating a commonly observed trend between fluctuation wavelength and relaxation time. (**e**) Autocorrelation amplitude (complex magnitude) of the spatial Fourier transform plotted as a function of time offset from a representative cell. Each line corresponds to a different spatial frequency in the range of 0.22–0.62 $\mu m^{-1}$ (corresponding to a wavelength in the range of 4.5–1.6 $\mu m$) in 0.056 $\mu m^{-1}$ intervals. (**f**) Best fit of the autocorrelation data shown in (**e**) to an exponential decay, fitted out to a drop in amplitude of 2/e. (**g–h**) Fitted parameters of the autocorrelation averaged over 67 cells. Data from this figure can be found in *Figure 1—source data 1*.

The online version of this article includes the following source data and figure supplement(s) for figure 1:

**Source data 1.** Source data corresponding to plots in *Figure 1*.

**Figure supplement 1.** Overview of cell segmentation analysis pipeline.

**Figure supplement 2.** Overview of analysis pipeline to extract fine-scale leading edge shape features.

**Figure supplement 3.** Control for analysis I.

**Figure supplement 4.** Control for analysis II.

**Figure supplement 5.** Control for analysis III.

**Figure supplement 6.** Leading edge fluctuation behavior is reproduced in fish epidermal keratocytes.

on leading edge curvature to sustain stable lamellipodial growth. These rich, measurable fine-scale dynamics therefore provide a unique opportunity to directly observe the time-evolution of leading edge maintenance. Taking advantage of our high-precision measurements, we aimed to quantitatively investigate the properties of the observed fluctuations, with the goal of determining the mechanisms

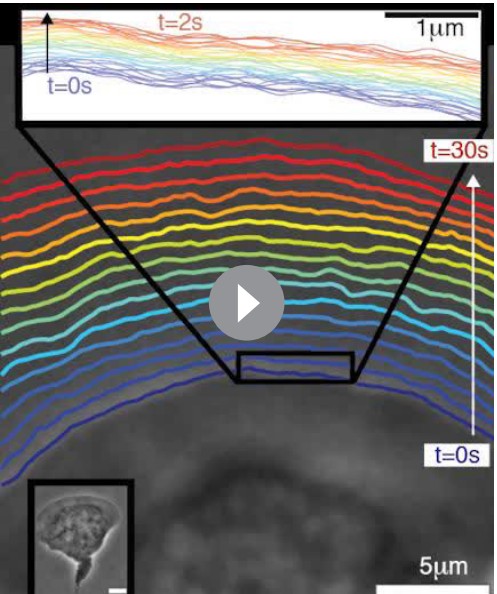

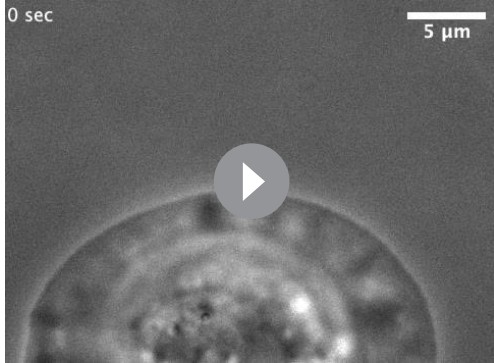

**Video 2.** Example fish epidermal keratocyte. Time lapse video corresponding to the data shown in Figure 1—figure supplement 6a-e.
https://elifesciences.org/articles/74389/figures#video2

**Video 1.** Segmentation overlaid onto migrating HL-60 cell. Time lapse video representation of segmentation results shown in Figure 1a.
https://elifesciences.org/articles/74389/figures#video1

by which molecular machinery at the leading edge coordinates the stochastic polymerization of individual actin filaments.

## Lamellipodial stability mediated by viscous relaxation of shape fluctuations

The relaxation of fine-scale shape fluctuations back to the steady-state leading edge shape is essential for the long time scale stability of lamellipodia. As for any physical system, the nature of this relaxation reflects the system's underlying physical properties; in this case, the characteristics of – and interactions between – actin filaments and the membrane. To provide a framework for exploration of the physical mechanisms underlying stable lamellipodial protrusion, we quantified the relaxation dynamics by performing time-autocorrelation analysis on the leading edge shape (Materials and methods). Applied in this context, this analytical technique calculates the extent to which the lamellipodium contour loses similarity with the shape at previous time points as fluctuations emerge and relax. As most material systems (actively-driven or otherwise) exhibit relaxation behavior with a characteristic wavelength-dependence (e.g. *Figure 1d*), we performed Fourier decomposition on the leading edge shape to separate out fluctuations at different length scales, and then performed autocorrelation analysis separately on each Fourier mode. We validated our analytical methods using simulations of membrane dynamics, for which there exists a well-established analytical theory (Materials and methods, *Figure 1—figure supplement 3*), and show that our results are not sensitive to an extension of our analysis to longer length and time scales (Materials and methods, *Figure 1—figure supplements 4–5*). Further, the membrane simulation control nicely demonstrates how visual features of curvature kymographs (e.g. *Figure 1b*) can be misleading (Materials and methods, *Figure 1—figure supplement 3*), and motivates the necessity of our more comprehensive technique.

Autocorrelation analysis revealed a monotonic relaxation of shape fluctuations at each wavelength (*Figure 1e–f*); the decay at every spatial scale is well-fit by an exponential form (*Figure 1f*), consistent with overdamped viscous relaxation. Importantly, we do not detect any increase in the autocorrelation over time, which would have appeared if there were any sustained, correlated growth of the fluctuations before they decay. This again suggests that the fluctuations arise from uncorrelated stochastic processes, such as fluctuations in actin density. A clear wavelength-dependence is observed, with shorter wavelengths decaying faster and having smaller amplitudes (*Figure 1g–h*). This general trend is shared by many physical systems with linear elastic constraints, such as idealized membranes (*Brown, 2008*) and polymers (*De Gennes, 2002*) freely fluctuating under Brownian motion, but can be contrasted with systems that have a dominant wavelength, as in the case of buckling or wrinkling of materials under compression (*Cerda and Mahadevan, 2003*). Importantly, these qualitative and quantitative properties of the leading edge fluctuations are not specific to cell type or experimental

conditions (e.g. agarose overlay, ECM), as we also observe this phenomenon in fish epidermal keratocytes (*Figure 1—figure supplement 6*, *Video 2*).

## Leading edge stability as an emergent property of branched actin growth

The rich behavior and quantitative nature of our leading edge shape fluctuation data made them ideal for comparison with physical models. In order to understand how molecular-scale actin assembly and biomechanics might give rise to the observed micron-scale shape dynamics, we aimed to reproduce this behavior in a stochastic model of branched actin network growth against a membrane (*Figure 2a–c*, *Video 3*, Materials and methods). Previous stochastic models of protrusive actin-based forces largely focused on actin polymerization against rigid obstacles (e.g. the bacterial cell wall for the *Listeria* comet tail *Carlsson, 2001*; *Carlsson, 2003* or a single, flat membrane segment in models of lamellipodia *Mueller et al., 2017*). Expanding on this general framework, and in an approach conceptually similar to previous work simulating small (< 2 µm) patches of a lamellipodium (*Schaus et al., 2007*; *Schaus and Borisy, 2008*), we incorporated a two-dimensional leading edge with filaments polymerizing against a flexible membrane, which we modeled as a system of flat membrane segments coupled elastically to each other. The size of the membrane segments was comparable to the spatial resolution of our experimental measurements, allowing us to assay fluctuations over a similar dynamic range of wavelengths. Simulated filaments apply force to the membrane following the classic untethered Brownian ratchet formalism (*Mogilner and Oster, 1996*), consistent with recent experiments showing that cellular protrusions are formed by largely untethered actin networks (*Bisaria et al., 2020*). Designed to be as comparable as possible to our experimental data, the model incorporated experimentally measured values from the literature for the membrane tension, membrane bending modulus, and biochemical rate constants (*Tables 1–2*, *Mogilner and Oster, 1996*; *Lieber et al., 2013*). As we were specifically interested in identifying biophysical mechanisms regulating leading edge stability, we minimized the model's biological complexity by including only the core biochemical elements of actin network growth dynamics: polymerization, depolymerization, branching, and capping. All filament nucleation in the model occurs through dendritic branching observed in cells to be mediated by the Arp2/3 complex (*Welch et al., 1997*; *Svitkina et al., 1997*), which catalyzes the nucleation of new 'daughter' actin filaments as branches from the sides of pre-existing 'mother' filaments at a characteristic angle of ~70° (*Mullins et al., 1998*; *Volkmann et al., 2001*; *Rouiller et al., 2008*). By simulating individual filament kinetics, the model captures the evolutionary dynamics of the filament network, allowing us to directly test hypothesized mechanisms for the interplay between actin network properties (e.g. filament orientation) and protrusion dynamics (*Mogilner and Oster, 1996*; *Lacayo et al., 2007*; *Mueller et al., 2017*; *Grimm et al., 2003*; *Maly and Borisy, 2001*; *Schaus et al., 2007*; *Figure 2b*).

To our great surprise, this very simple model was able to recapitulate stable leading edge fluctuations. Nascent leading edges reach steady state values for filament density, filament length, filament angle, membrane velocity, and (most importantly) membrane fluctuation amplitude within seconds, a biologically realistic time scale (*Figure 2d–h*). Furthermore, the steady state values obtained are in quantitative agreement with both our own experimental data and previously published measurements, with the model yielding mean values of: 0.3 filaments/nm for filament density (~30 nm filament spacing for a lamellipodium that is 10 filaments tall) (*Abraham et al., 1999*), ~150 nm for filament length, ~40° for filament angle (with respect to the direction of migration), ~0.35 nm/ms for membrane velocity, and ~50 nm for membrane fluctuation amplitude (*Svitkina et al., 1997*; *Maly and Borisy, 2001*; *Verkhovsky et al., 2003*).

As observed in the experimental measurements, simulated leading edge shape stability is mediated by an exponential decay of shape fluctuations (*Figure 2i–j*). Furthermore, the minimal model correctly predicts the monotonic trends of fluctuation amplitude and decay time scale with wavelength (*Figure 2k–l*) in a way that was not sensitive to our choices of simulation time step, membrane segment length, and overall length of the leading edge (*Figure 2—figure supplements 1–3*). It should be noted that the generation of the simulated data in *Figure 2k–l* did not involve any curve-fitting (and therefore *no free parameters* that could be fit) to the experimentally measured autocorrelation dynamics in order to parameterize the model. Rather, the simulated fluctuation relaxation behavior, qualitatively reproducing our experimental measurements, emerges directly from the molecular-scale

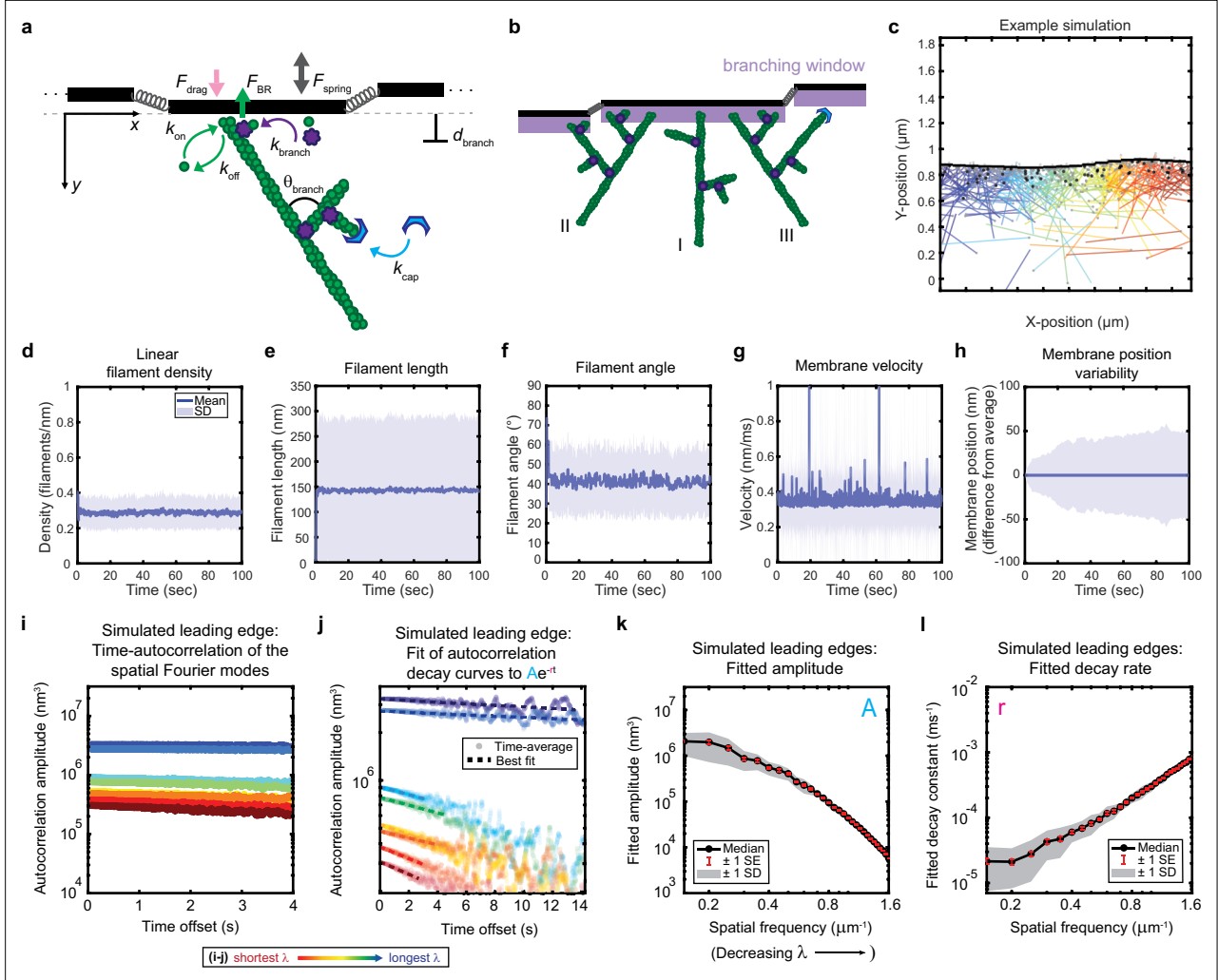

**Figure 2.** Minimal model of branched actin growth recapitulates leading edge stability and shape fluctuation relaxation. (**a–b**) Model schematic. Black lines, membrane; green circles, actin; purple flowers, Arp2/3 complex; blue crescents, capping protein. Rates: $k_{on}$, polymerization; $k_{off}$, depolymerization; $k_{branch}$, branching; $k_{cap}$, capping. $\theta_{branch}$, branching angle. $d_{branch}$, branching window. Physical parameters: $F_{spring}$, forces between membrane segments; $F_{BR}$, force of filaments on the membrane (Brownian ratchet); $F_{drag}$, viscous drag. (**b**) Schematic demonstrating filament angle evolution. Filaments growing perpendicular to the leading edge (I) outcompete their progeny (branches), leading to a reduction in filament density; filaments growing at an angle (II and III) make successful progeny. Filaments spreading down a membrane positional gradient (II) are more evolutionarily successful than those spreading up (III). (**c**) Simulation snapshot: Black lines, membrane; colored lines, filament equilibrium position and shape; gray dots, barbed ends; black dots, capped ends; filament color, x-position of membrane segment filament is pushing (increasing across the x-axis from blue to red). (**d–h**) For a representative simulation, mean (solid line) and standard deviation (shading) of various membrane and actin filament properties as a function of simulation time. Note for linear filament density (**d**) lamellipodia are ~10 filament stacks tall along the z-axis, giving mean filament spacing of 10/density, or ~30 nm. (**i–l**) Autocorrelation analysis and fitting for a representative simulation (**i–j**) as well as best fit parameters averaged over 40 simulations (**k–l**). Data from this figure can be found in *Figure 2—source data 1*.

The online version of this article includes the following source data and figure supplement(s) for figure 2:

**Source data 1.** Source data corresponding to plots in *Figure 2*.

**Figure supplement 1.** Simulations are performed at sufficient temporal discretization.

**Figure supplement 2.** Simulations are performed at sufficient spatial discretization.

**Figure supplement 3.** Simulated leading edge behavior is not affected by leading edge length.

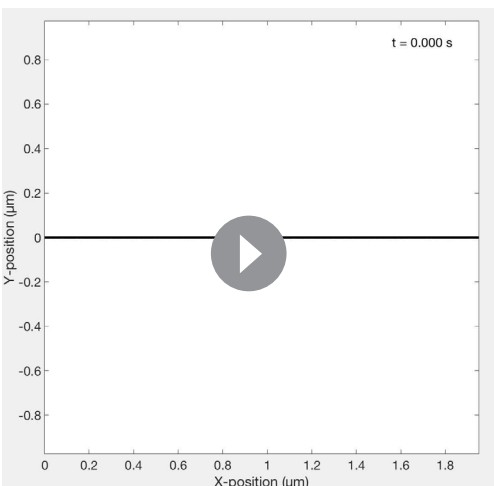

**Video 3.** Example simulation. Time lapse video representation of simulation results shown in Figure 2c. https://elifesciences.org/articles/74389/figures#video3

actin growth model, in which all biochemical parameters were estimated from measurements in the existing literature (*Tables 1–2*) – leaving no free simulation parameters.

## Predicting effects of drug treatment with Latrunculin B

We were interested in further assaying the predictive power of this minimal stochastic model by determining whether the output of the simulations was congruent with experimental observations under conditions that had not been tested prior to model development. As an example, we elected to test whether the model could correctly predict the response of HL-60 cells to treatment with the drug Latrunculin B, which binds to and sequesters actin monomers. Qualitatively, cells treated with Latrunculin B (*Video 4*) present with enhanced bleb formation and more variable leading edge shapes, in comparison with cells treated with a DMSO vehicle control (*Video 5*).

In our model, addition of this drug can be simulated by reducing the free monomer concentration, which consequently reduces both the polymerization rate and the branching rate. At low effective doses, subtle but measurable changes to leading edge fluctuations were predicted: specifically, an increase in the amplitudes at large wavelengths, and a decrease in the decay rates across all wavelengths (*Figure 3a–b*). Our experimental results were consistent with these quantitative predictions; Latrunculin B-treated cells exhibited increased fluctuation amplitudes and decreased fluctuation rates over the predicted ranges (*Figure 3c–d*).

## Geometry as the core determinant of simulated leading edge stability

Given the success of the model in reproducing experimental results, we next wanted to determine which features of the simulation were responsible for leading edge stability and relaxation of fluctuations. The simplicity of the model allowed us to determine the stability mechanism by process of elimination, selectively removing elements of the model (Materials and methods) and determining whether stability was retained. To assay the importance of membrane tension and bending rigidity, which has been suggested to be a key factor regulating lamellipodial organization (*Batchelder et al., 2011*; *Sens and Plastino, 2015*), we simply removed the forces between the membrane segments (*Figure 2a*, $F_{spring}$) from the simulation (Materials and methods). Surprisingly, the coupling between the membrane segments (i.e. the effects of tension and bending at length scales larger than the size of an individual membrane segment) was completely dispensable for leading edge stability (*Figure 4a–e*).

Following a similar process of elimination, we determined that in fact only two elements were required for stability. First, as reported previously for dendritic actin network polymerization against a single stiff obstacle, it was necessary to constrain branching to occur only within a fixed distance from the leading edge membrane (*Figure 2a–b*) in order to maintain a steady state actin density (*Carlsson, 2001*). The molecular motivation for this spatially limited 'branching window' is rooted in that fact that activators of the Arp2/3 complex, which render Arp2/3 competent for actin filament nucleation, are typically membrane-associated proteins (*Suetsugu, 2013*). Second, we found that stability is inherently tied to the ability of filaments to spread laterally to neighboring membrane segments (*Figure 2b* II-III, *Figure 4f–i*). Recall that, because the branched actin network geometry causes filaments to grow, on average, at an angle relative to the membrane normal (*Maly and Borisy, 2001*; *Verkhovsky et al., 2003*), polymerizing tips spread laterally along the leading edge (*Lacayo et al., 2007*; *Grimm et al., 2003*). Removing filament spreading from the model by fixing filaments to remain associated with their nearest membrane segment at birth (Materials and methods) led to actin density divergence: network regions with low filament density eventually underwent complete depolymerization, while high-density regions continued to accumulate actin (*Figure 4f–g*).

**Table 1.** Actin network growth parameters.

Parameters listed are the default used for the simulations.

| Notation | Meaning | Value | Source |
|---|---|---|---|
| $M$ | Free monomer concentration | 15 µM | *Cooper, 1991*; *Marchand et al., 1995* |
| $k_{on}$ | Polymerization rate | $11 \cdot 10^{-3}$ monomers ms$^{-1}$ µM$^{-1}$ | *Pollard, 1986* |
| $k_{off}$ | Depolymerization rate | $10^{-3}$ monomers ms$^{-1}$ | *Pollard, 1986* |
| $k_{cap}$ | Capping rate | $3 \cdot 10^{-3}$ ms$^{-1}$ | $\sim 3 \cdot 10^{-3}$ µM$^{-1}$ ms$^{-1}$ *Schafer et al., 1996* at 1 µM capping protein *Pollard et al., 2000* |
| $k_{branch}$ | Branching rate | $4.5 \cdot 10^{-5}$ branches ms$^{-1}$ µM$^{-1}$ nm$^{-1}$ | 50 nm branch spacing *Svitkina et al., 1997*; *Svitkina and Borisy, 1999*; Branch rate approximated such that elongation rate / branch rate = 50 nm; $k_{branch} = (k_{on} \cdot M \cdot l_m)/(50 \text{ nm} \cdot M \cdot y_{branch})$ |
| $y_{branch}$ | Branching window length | 15 nm | $\sim 3$–5 protein diameters away from the membrane |
| $\theta_{branch}$ | Branching angle | $70 \pm 10°$ | *Mullins et al., 1998*; *Volkmann et al., 2001*; *Rouiller et al., 2008*; *Blanchoin et al., 2000*; *Cai et al., 2008*; *Svitkina and Borisy, 1999* |
| $l_p$ | Actin filament persistence length | 1 µm | *Käs et al., 1996* |
| $l_m$ | Actin monomer length | 2.7 nm | *Pollard, 1986* |

These findings lead us to a simple molecular feedback mechanism for leading edge stability, based on a synergy between filament spreading and membrane-proximal branching (*Figure 4j*): To begin with, regions with initially high filament density come to protrude beyond the average position of the rest of the membrane, representing the emergence of a leading edge shape fluctuation (*Figure 4j*, I, II). This induces asymmetric filament spreading, where filaments from high-density regions can spread productively into neighboring regions (*Figure 2b*, II), but filaments spreading from adjacent low-density regions cannot keep up with the fast moving membrane segments (*Figure 2b*, III), and thus are unproductive (*Figure 4j*, II, III). In this way, the branched geometry inherent to dendritic actin polymerization, as well as its interaction with the shape of the membrane, naturally encodes leading edge stability (*Figure 4j*, IV). Thus our results directly demonstrate a 'stability-through-spreading' mechanism that has previously only been assumed in mean-field analytical theories (*Lacayo et al., 2007*; *Grimm et al., 2003*). Remarkably, this means that leading edge maintenance is an intrinsic, emergent property of branched actin network growth against a membrane, without requiring any further regulatory governance. Geometrical constraints imposed simply by the nature of membrane-proximal actin branching ensure that any small variations in *either* local actin filament density *or* growth rate are inherently self-correcting to regress toward the mean.

Of note, it has previously been shown that Arp2/3-mediated branching is required for lamellipodial formation in a wide variety of cell types; cells with inhibited or depleted Arp2/3 complex exhibit

**Table 2.** Physical parameters.

Parameters listed are the default used for the simulations.

| Notation | Meaning | Value | Source |
|---|---|---|---|
| $k_B$ | Boltzmann constant | 0.0183 pN nm K$^{-1}$ | – |
| $T$ | Temperature | 310.15 K | – |
| $\sigma$ | Membrane tension | 0.03 pN nm$^{-1}$ | *Lieber et al., 2013* |
| $\kappa$ | Membrane bending modulus | 140 pN nm | *Lieber et al., 2013* |
| $\eta_w$ | Viscosity of water at 37 °C | $7 \cdot 10^{-7}$ pN ns nm$^{-2}$ | – |
| $\eta$ | Effective viscosity at the leading edge | 3000 $\eta_w$ | $\sim$ effective viscosity of micron-scale beads in cytoplasm *Wirtz, 2009* |
| $L$ | Leading edge length | 20 µm | This work |
| $h$ | Leading edge height | 200 nm | *Abraham et al., 1999*; *Laurent et al., 2005*; *Urban et al., 2010* |
| $\Delta x$ | Membrane segment length | 100 nm | – |
| $N$ | Number of membrane segments | 200 | – |

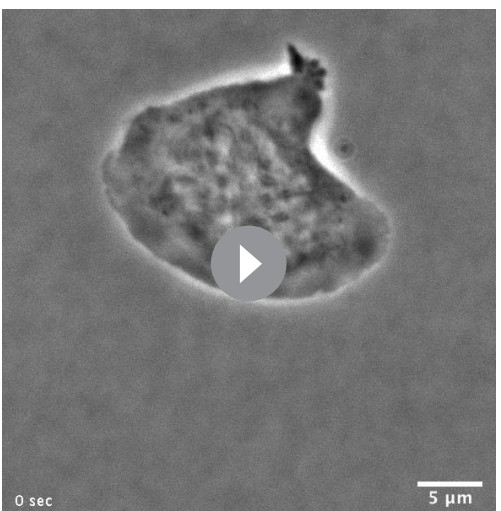

**Video 4.** Example HL-60 cell treated with 30 nM latrunculin B.

https://elifesciences.org/articles/74389/figures#video4

complete disruption of the lamellipodium shape and often switch to a different mode of migration altogether, such as filopodial motility (*Fritz-Laylin et al., 2017b*; *Henson et al., 2015*; *Wu et al., 2012*; *Davidson et al., 2018*). Indeed, HL-60s treated with the Arp2/3 inhibitor CK-666 have extremely variable leading edge shapes, characterized by long, thin filopodia-like protrusions (*Video 6*). Our theoretical results provide a mechanistic interpretation for this striking phenomenon, suggesting that the vital lamellipodial maintenance role of Arp2/3-mediated branching stems from its ability to mediate efficient filament spreading and equilibration of actin density fluctuations, purely because the daughter filament always grows at an angle distinct from its mother.

## Optimal suppression of fluctuations by the highly conserved ~70° branching angle

Given the essential contribution of branched network geometry to the stability of the simulated leading edges, we reasoned that variations in the branching geometry alone might have a significant effect on leading edge fluctuations. We therefore performed simulations to determine the effects of changing the average branching angle and branching angle variability on filament orientation, filament density, and leading edge fluctuation fit parameters (*Figure 5*). In this context, we highlight the distinction between the branching angle, $\theta_{br}$ (i.e. the angle of a daughter filament relative to its mother), and the filament angle or orientation, $\theta_f$ (i.e. the angle of a filament relative to the direction of migration) (*Figure 5a*, inset). Due to the sterotypical branching angle, $\theta_{br}$, there is a direct correspondence between the orientation, $\theta_f^{mother}$, of a mother filament and the orientation, $\theta_f^{daughter}$, she passes on to her daughter branches. Our simulations are thus, in effect, selection assays, as mother filaments compete to stay within the fixed branching window, spawn daughter branches, and thus pass down their angle to their progeny (*Maly and Borisy, 2001*; *Schaus et al., 2007*). For example, when filaments are initialized with a random orientation, and the branching angle is fixed (i.e. there is no variability in the branching angle), only a handful of the initial filament angles ($\theta_f$) survive until the end of the simulation (*Figure 5a*). The surviving, successful filament angles are narrowly and symmetrically distributed around one half of the branching angle (*Figure 5a–c*). This optimal filament angle allows mother and daughter filaments to branch back and forth symmetrically about the membrane normal, such that mother filaments do not out-compete their progeny (as has been described previously) (*Maly and Borisy, 2001*; *Schaus et al., 2007*; *Schaus and Borisy, 2008*; *Figure 2b*).

In living cells, branching is mediated by the Arp2/3 complex, which has been experimentally measured to form highly regular and sterotyped branches at ~70° (*Mullins et al., 1998*; *Volkmann et al., 2001*; *Rouiller et al., 2008*). Intriguingly, this protein complex is highly conserved (*Welch et al., 1997*), with measurements of the branching angle in a wide variety of species, including protists (*Mullins et al., 1998*; *Volkmann et al., 2001*; *Rouiller et al., 2008*; *Blanchoin et al., 2000*), yeast (*Rouiller et al., 2008*), mammals (*Rouiller et al., 2008*; *Blanchoin et al.,*

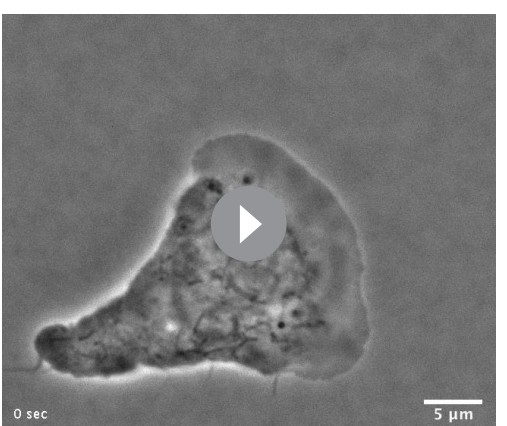

**Video 5.** Example HL-60 cell treated with 0.1% DMSO vehicle control.

https://elifesciences.org/articles/74389/figures#video5

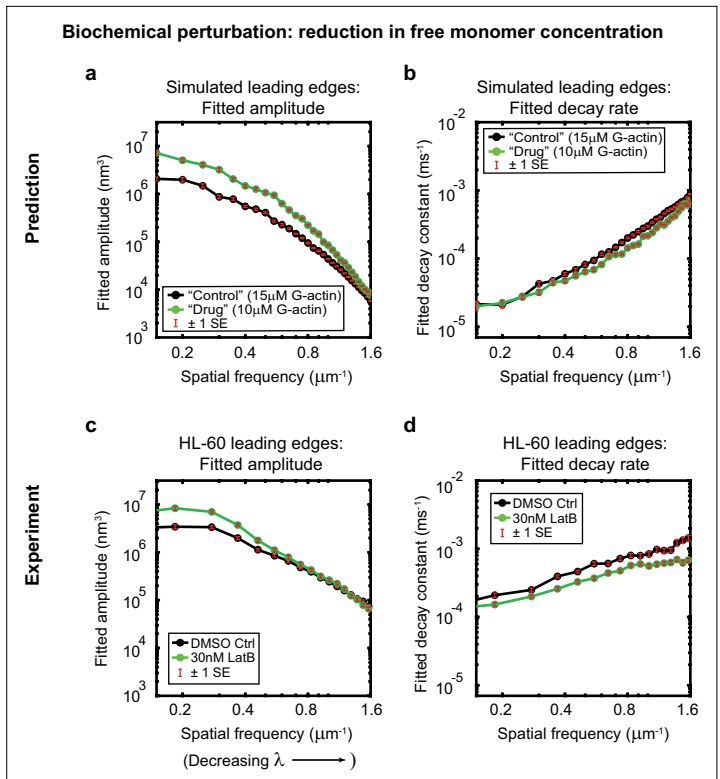

**Figure 3.** Minimal model correctly predicts response of HL-60 cells to drug treatment. Predicted and experimentally-measured response of the autocorrelation decay fit parameters to drug treatment with Latrunculin B, plotted as in *Figure 1g–h*. (**a–b**) Predicted response to a reduction in the free monomer concentration (green, 10 μM G-actin) compared to the standard concentration used in this work (black, 15 μM G-actin) – medians over 40 simulations for each condition. (**c–d**) Experimentally measured behavior: DMSO control – medians over 67 cells (same data as plotted in *Figure 1g–h*). 30 nM Latrunculin B – medians over 34 cells. Data from this figure can be found in *Figure 3—source data 1*.

The online version of this article includes the following source data for figure 3:

**Source data 1.** Source data corresponding to plots in *Figure 3*.

---

*2000*; *Cai et al., 2008*), and amphibians (*Svitkina and Borisy, 1999*), using various experimental techniques (platinum replica electron microscopy, cryo-electron microscopy, and total internal reflection microscopy), all falling within the range of 67–78° (± 2–13°). The high degree of conservation hints that this specific angle might carry some functional optimality, but the question has not been addressable experimentally; due to the lack of naturally occuring Arp2/3 variants with a substantially different branching angle, an alternative branching structure would hypothetically have to be designed do novo, presumably by altering the protein interaction interface by which the Arp2/3 complex binds to the side of a mother filament (*Volkmann et al., 2001*). We thus sought to explore the possible functional significance of this conserved angle using our minimal stochastic model. Excitingly, we found that in simulations with no branch angle variability, a 70–80° branching angle was optimal for minimizing both actin density fluctuations (*Figure 5d and i*) and leading edge fluctuation amplitudes for wavelengths smaller than ~2 μm (*Figure 5f and h*). These smoothing effects are therefore predicted to be relevant within the experimentally-measurable range of wavelengths (between ~0.7 μm and ~2 μm), but are most beneficial for the smallest wavelengths resolved by our simulations (down to ~0.3 μm) – closest to the length scales of individual filament polymerization. Overall, these results provide tantalizing mechanistic insight into the long-standing question of why the characteristic branching angle is so ubiquitous.

Heritability of filament orientation (i.e., the extent to which mother filament orientation determines the orientation of the daughters) is set by the degree of variability in the branching angle (which, in turn, reflects the influence of thermal fluctuations). Perhaps unsurprisingly, decreasing this

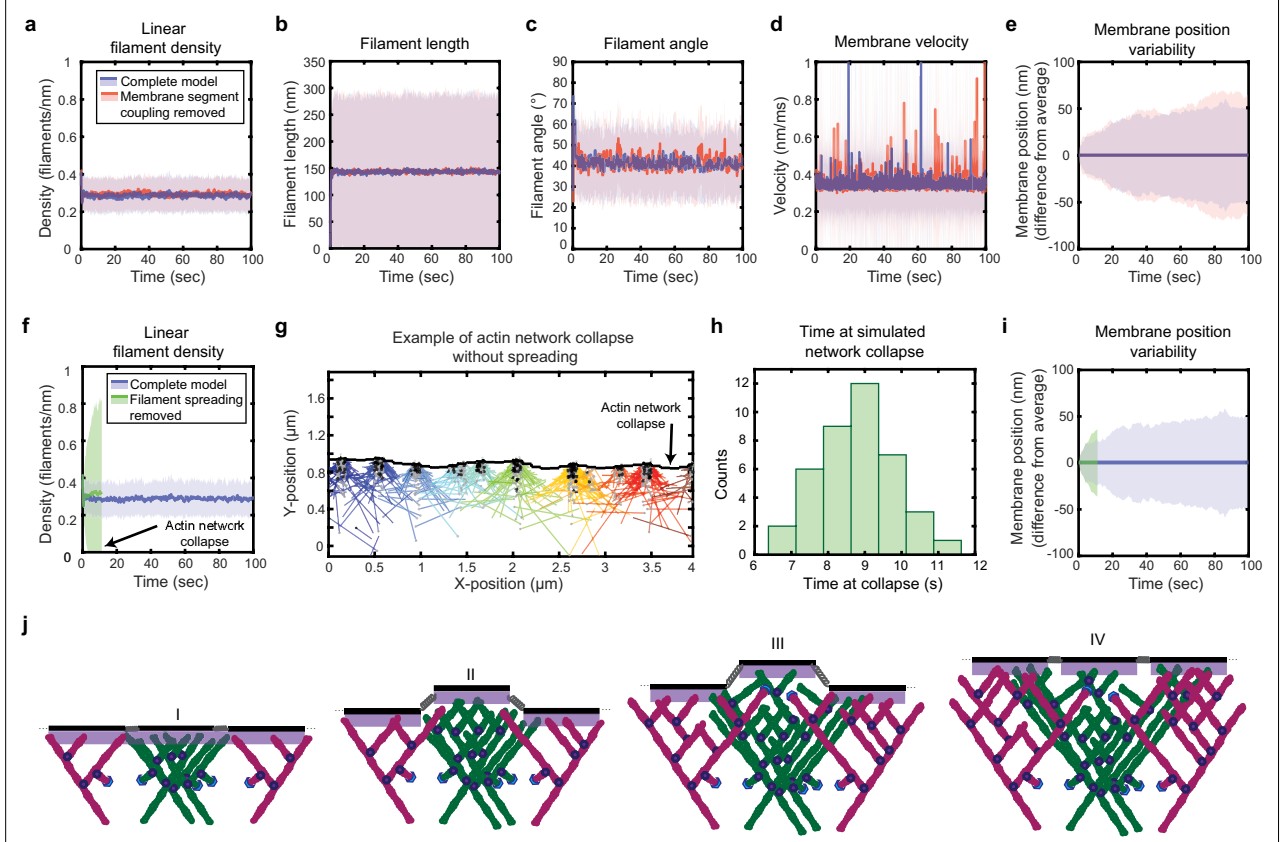

**Figure 4.** Simulated lamellipodial stability is governed by leading edge geometry. (**a–e**) Comparison of leading edge properties with and without the coupling of the membrane segments by tension and bending rigidity (no coupling: $F_{spring}$ = 0 in *Figure 2a*), plotted as in *Figure 2d–h*. (**f–i**) Comparison of leading edge properties with and without the ability of filaments to spread between neighboring membrane segments. (**g**) A snapshot of the simulation after filament network collapse (defined as a state where at least 25% of the membrane segments have no associated filaments). (**f,g,i**) Plots made from the same simulation. (**h**) A histogram of the average time to network collapse over 40 simulations. (**j**) Schematic representing the proposed molecular mechanism underlying the stability of leading edge shape, with time increasing from I-IV. Data from this figure can be found in *Figure 4—source data 1*.

The online version of this article includes the following source data for figure 4:

**Source data 1.** Source data corresponding to plots in *Figure 4*.

orientational heritability significantly reduces the dependence of fluctuation amplitude and filament density variability on the branching angle (*Figure 5j–q*), and thereby counteracts the beneficial effect of the optimal angle on leading edge fluctuations. Introducing a branch angle variability of ±2° (on the lower end of the experimentally-measured values) broadens the range of near-optimal branch angles but maintains the optimum at ~70–80° (*Figure 5j and l*), while introducing a variability of ±10° (on the higher end of the measured range) completely removes the optimum (*Figure 5n and p*). In both cases, increasing the branch angle variability increases the minimum possible fluctuation amplitude (*Figure 5h, l and p* – insets) and filament density variability (*Figure 5i, m and q*), representing a decrease in the noise-suppression capabilities of the system. Overall, these results provide strong support for the idea that actin network geometry is not only essential for leading edge stability, but also plays a major role in determining the fundamental limits of smoothness in lamellipodial shape.

## Discussion

The emergence of robust collective behaviors from stochastic elements is an enduring biological mystery which we are only beginning to unravel (*Huang et al., 2016*; *Battich et al., 2015*; *Chang and Marshall, 2017*; *Mohapatra et al., 2016*; *Raj and van Oudenaarden, 2008*; *Gray et al., 2019*; *Oates, 2011*). The apparent dichotomy in actin-based motility between the random elongation of

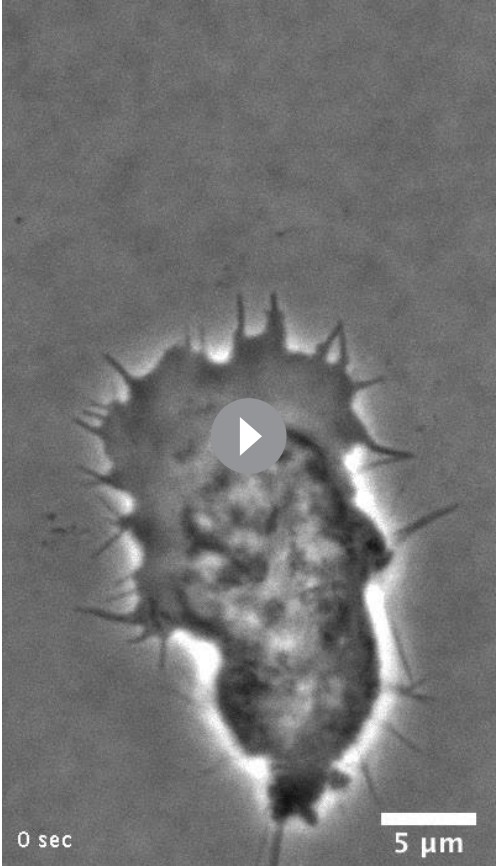

**Video 6.** Example HL-60 cell treated with 100 µM CK-666.

https://elifesciences.org/articles/74389/figures#video6

individual filaments and the stable formation of smooth and persistent higher order actin structures such as lamellipodia exemplifies this enigma, and provides an avenue toward understanding general strategies for noise suppression in biological systems.

In recent years it has become clear that perturbation-free experiments which examine fluctuations around the mean at steady state (in contrast to probing the change in the mean due to a perturbation) can be a powerful tool for understanding noisy systems (*Welf and Danuser, 2014*). In this work, application of that principle in combination with high-precision measurements, quantitative analytical techniques, and physical modeling led to the surprising revelation that the suppression of stochastic fluctuations *naturally emerges* from the interactions between a growing actin network and the leading edge membrane, with *no additional feedback required*. Our insights into the molecular mechanisms mediating lamellipodial stability were largely enabled by experimentally measuring micron-scale leading edge shape dynamics and comparing them to a molecular-scale actin network growth model that correctly predicts this emergent behavior (as well as many other experimentally-measured features of lamellipodial actin networks). Ultimately, we hope our results inspire future experimental work to directly measure the nanometer-scale interactions predicted by our simulations, which may be accomplished using super-resolution imaging of actin dynamics in vivo or in an in vitro reconstitution of lamellipodial protrusion (i.e. branched polymerization of actin driving the motion of a flexible barrier).

The model developed in this work provides an understanding of the basic biophysical mechanisms underlying lamellipodial migration. Of course, living cells are home to array of additional complexities which are likely to further modulate leading edge fluctuations and stability, and our model may provide a framework for future exploration of such effects across diverse cell types and experimental conditions. Cells migrating in vivo inevitably experience a much more challenging and dynamic environment, in which additional feedback mechanisms will almost certainly be required for the maintenance of polarized migration. The simplicity and biophysical realism of our modeling framework should make it particularly well-suited for future studies focused on predicting and understanding the effects of additional potential feedback mechanisms, including tethering (*Mogilner and Oster, 2003*; *Soo and Theriot, 2005*; *Alberts and Odell, 2004*; *Kuo and McGrath, 2000*), a limiting monomer concentration (*Mullins et al., 2018*), force-dependent branching (*Risca et al., 2012*; *Parekh et al., 2005*; *Chaudhuri et al., 2007*), and regulation by curvature-sensing proteins (*Tsujita et al., 2015*; *Zhao et al., 2011*). Further, this computational model might be useful for exploring the effects of various extracellular forces, such as those produced by obstacles or variations in matrix density, or intracellular forces, such as those produced by hydrostatics. We also note that a certain degree of biochemical signaling is implicit in our model in the form of biochemical rate constants that are invariant in time and space; this property relies on signaling networks to maintain uniform gradients of actin-associated molecules (*Devreotes et al., 2017*). How local biochemical control (or lack thereof) over these rate constants might affect leading edge fluctuations remains an interesting avenue for future investigation, both theoretically and experimentally.

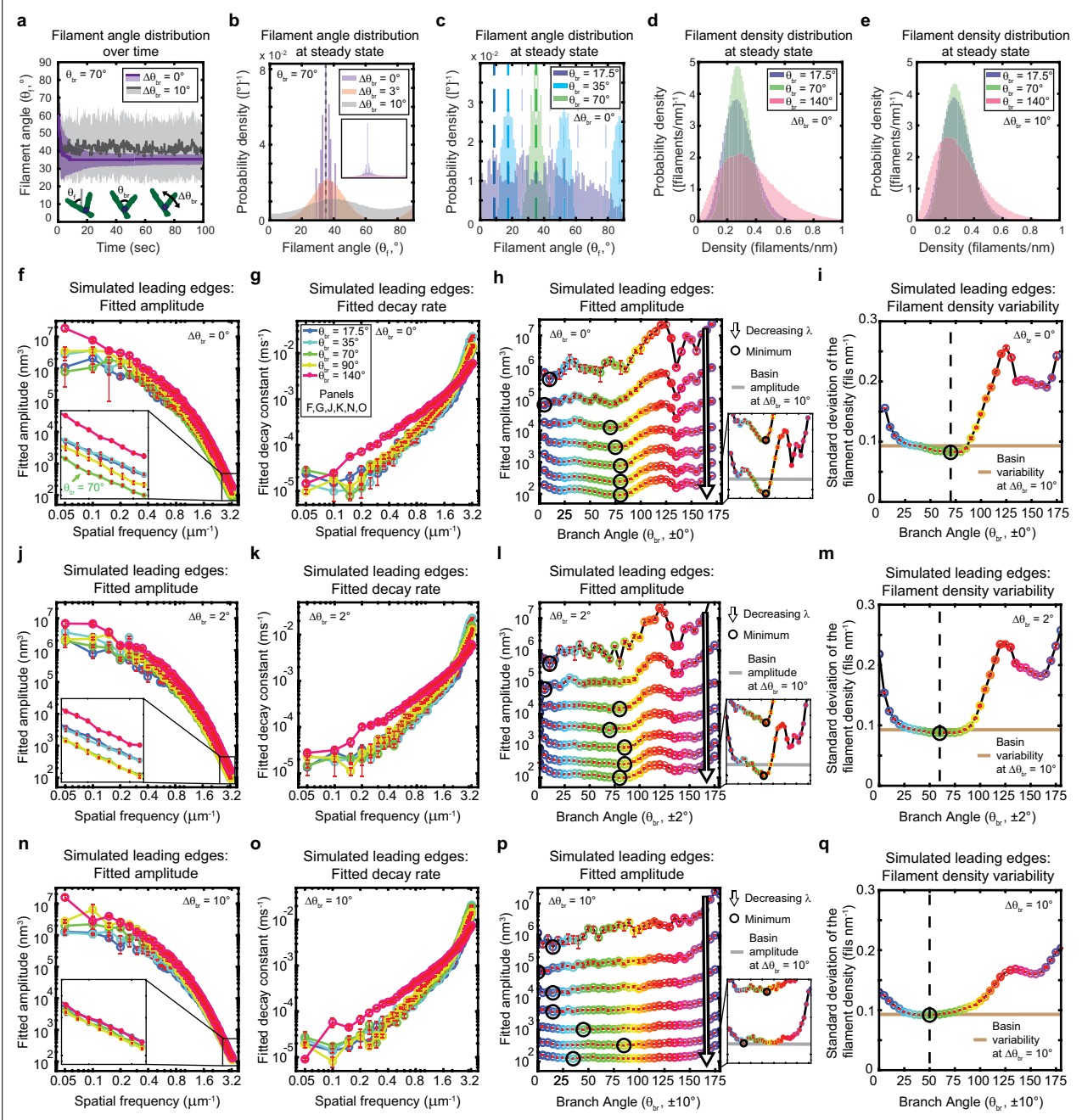

**Figure 5.** The genetically-encoded Arp2/3-mediated branching angle is optimal for suppressing leading edge fluctuations. (a–c) Time course (a) and steady state distribution (b–c) of the filament angle ($\theta_f$) for simulations with various branching angle standard deviations ($\Delta\theta_{br}$), (a–b) and means ($\theta_{br}$), (c). Dashed lines represent $\theta_{br}/2$ plus integer multiples of $\theta_{br}$. (d–e) Steady state filament density distribution as a function of the mean branching angle in the context of $\Delta\theta_{br}$=0° (d) and $\Delta\theta_{br}$=10° (e). (a–b, d–e) Results from a representative simulation for each condition. (c) Data integrated over 40 simulations. (f–q) Predicted response of leading edge fluctuations and filament density variability to changes in the branch angle and branch angle variability, medians over 40 simulations for each condition. Red error bars – standard error. Color map is identical for panels c-q. (h,l,p) Fitted amplitude as a function of branch angle, where each line represents a different spatial frequency, increasing from 0.2 to 3.2 µm⁻¹ in intervals of 0.5 µm⁻¹. Insets have identical x-axes to main panels. (i,m,q) Standard deviation of the filament density at steady state plotted as a function of branch angle. Note the x- and y-axes limits in (f–g, j–k, n–o) are expanded compared to the equivalent panels in *Figures 1–3*. Data from this figure can be found in *Figure 5—source data 1*.

The online version of this article includes the following source data for figure 5:

**Source data 1.** Source data corresponding to plots in *Figure 5*.

The defining characteristic and major advance of our model was the explicit inclusion of both the evolutionary dynamics of the actin network and its interaction with the two-dimensional geometry of the leading edge. By selectively removing elements of the model, we determined that lateral filament spreading, combined with a fixed branching window, is indispensable for leading edge stability. This highlights the crucial role in lamellipodial maintenance of the branched structure of actin networks, wherein each daughter filament inherits angular information from its mother. Our further investigations into the evolutionary properties of actin network growth revealed that a ~70–80° branch angle maximally suppresses fine-scale actin density and leading edge shape fluctuations, showing for the first time that Arp2/3-mediated branching imparts optimal functionality, as was long hypothesized based on strong sequence, structural, and functional conservation throughout the eukaryotic tree of life. It is interesting to note that the evolutionarily-conserved branching angle that we find maximally suppresses leading edge fluctuations appears not to be the same angle that optimizes the polymerization velocity of single filaments – predicted to be a broad angle closer to ~90–100° (and quite load-dependent) in the low-load regime (*Mogilner and Oster, 1996*). This contrast suggests that evolutionary selection acts at the level of actin network properties, rather than force production by individual filaments.

Returning to the broader question of how noisy biological systems control for stochasticity, we find that stability in the case of lamellipodial dynamics is inherently encoded by the geometry of branched actin network growth. It will be interesting to see whether similar principles hold for other cytoskeletal structures with clear geometric constraints, such as endocytic pits, the cytokinetic ring, and the mitotic spindle.

## Materials and methods

### HL-60 cell culture and differentiation

HL-60 cells were cultured as described previously (*Millius and Weiner, 2009*; *Garner et al., 2020*). In brief, cells were maintained at a density of 0.1–1 x $10^5$ cells/mL by passaging every 2–3 days into fresh RPMI media supplemented with 10% heat-inactivated fetal bovine serum and antibiotics/antimycotics. Supplementation with 1.57% DMSO was used to differentiate the cells into a neutrophil-like state. Cells were subsequently extracted for experiments at 6 days post-differentiation. Our HL-60 cell line was obtained from Orion Weiner's lab at UCSF, who originally received them from Henry Bourne's lab at UCSF. The identity of this suspension cell line was confirmed based on the behavior of the cells, including differentiating into a neutrophil-like state upon exposure to DMSO that exhibits characteristic phenotypes for substrate adhesion, rapid migration, and elongated morphology. HL-60s are not listed as a misidentified cell line on the Register of Misidentified Cell Lines. The cell line tested negative to mycoplasma contamination.

### Under-agarose motility assays with HL-60s

Differentiated HL-60 cells were plated on fibronectin-coated coverslips and then overlaid with a 1% agarose pad containing 1 nM fMLP (to enhance migratory behavior), as described previously (*Garner et al., 2020*). Microscopy of the migrating cells was performed at 37 °C, using transmitted light to image phase contrast on an epifluorescence microscope at ×100 magnification (100 × 1.45 NA Plan Apo oil objective, Nikon MRD31905). A more detailed description of our microscopy system can be found in previous publications (*Garner et al., 2020*). For treatment with Latrunculin B or CK-666, the drug was embedded into the agarose pad by adding the drug to the unpolymerized agarose pad solution before gelling (at the same time as adding fMLP), such that drug treatment begins when cells are overlaid with the agarose pad and is maintained throughout imaging. Cells were imaged at 45 min post-plating. Drugs were first diluted down to 1000 X in DMSO, then added to the agarose solution at a dilution of 1:1,000 (for a final concentration of 30 nM for Latrunculin B and 100 µM for CK-666), giving a final DMSO concentration in the pad of 0.1%. Controls were performed by adding 0.1% DMSO to the agarose pad alone.

### Keratocyte isolation and motility assays

Keratocytes were cultured from wild-type zebrafish embryos at 2 days post-fertilization as described previously (*Lou et al., 2015*). Briefly, zebrafish embryos were collected at 2 days post-fertilization,

dechorionated, and anesthetized using tricaine. To dissociate the keratocytes, dechorionated fish were then washed in PBS, incubated in Cell Dissociation Buffer for 30 min at 4 °C, incubated in a solution of 0.25% trypsin and 1 mM EDTA for ~15 min at 28 °C, and then incubated in fetal bovine serum to quench the trypsin. From this point on, cells were maintained in antibiotic and antimycotic to deter microbial growth. The keratocyte-rich supernatant was then concentrated by centrifugation at 500 g for 3 min. Keratocytes were then plated on collagen-coated coverslips and incubated at room temperature for ~1 hr to allow cells to adhere. Once adherent, the supernatant was exchanged for imaging media (10% fetal bovine serum in L-15) and allowed to incubate another 15 min at room temperature before imaging. The keratocytes were imaged at 28 °C under similar conditions as those used for HL-60 cells. Data included in *Figure 1—figure supplement 6* represent cells from a single coverslip. Experiments were approved by University of Washington Institutional Animal Care and Use Committee (protocol 4427–01).

## Image segmentation

Most segmentation algorithms penalize curvature in the contour in order to reduce the noise in the fitting algorithm (*Seroussi et al., 2012*). However, this runs the risk of introducing artificial correlations and structure into the data. For example, a spring-like curvature penalty would artificially make fluctuations appear to be stretch-dominated. Therefore, we performed the following custom segmentation algorithm to avoid these potential artifacts (*Figure 1—figure supplement 1*). Phase contrast time-lapse videos were manually aligned to the direction of motion of the cell, such that each cell migrates up the y-axis on an x-y coordinate system (*Figure 1—figure supplement 1a, b*). The videos were then cropped to isolate the cell leading edges and exclude the cell body, for easier segmentation. If the cell migrates up the y-axis, this means every image pixel along the x-axis has an associated leading edge position along the y-axis. A leading edge y-position was assigned to each x-axis pixel independently of knowledge about neighboring x-axis pixels, to avoid injecting the artifacts discussed above. A manual segmentation was performed for the first time point in each movie. A custom, automated segmentation algorithm written in MATLAB then performed a line scan of the phase contrast intensity along the y-axis separately for each pixel along the x-axis. For each line scan, the algorithm performs a local search for the leading edge position, constrained to be within a fixed number of pixels from either the manual segmentation (for the first time point) or the previous time point (for subsequent timepoints). The leading edge position is defined as the midpoint between the brightest phase intensity (phase halo) and the point of steepest intensity gradient (transition from phase halo to phase-dense cytoplasm).

## Preparation of the curvature and velocity kymographs

The curvature and velocity kymographs (*Figure 1b–c*, *Figure 1—figure supplement 3*) were prepared using custom MATLAB code. Curvatures are calculated as the inverse radius of the best-fit circle corresponding to a 30 pixel-wide (~1.5 µm) region about each position. The most prominent fluctuation events seen in the curvature kymographs (*Figure 1b*, *Figure 1—figure supplement 3*) somewhat correspond to (but do not exactly match) the length of the fitting window. For example, the simulated and experimental data shown in *Figure 1—figure supplement 3* were fit using the same ~1.5 µm fitting window and have similar apparent 'dominant wavemodes' of ~3 µm, or twice the fitting window. However, despite being fit with the same fitting window, the simulated data has an observably smaller apparent 'dominant wavemode' than the experimental data. Velocities were calculated as the distance traveled over 250ms (five 50ms timepoints) non-overlapping windows.

## Processing of segmented cell shapes for autocorrelation analysis

Kymographs of curvature and velocity such as those shown in *Figure 1b–c*, while helpful to obtain a qualitative sense of the fluctuation data, are visually dominated by the largest size-scale features of the leading edge. They thus offer an incomplete description of the shape fluctuations (*Ma et al., 2018*) – notably de-emphasizing the fine-scale features that are the subject of this study. Further, curvature kymographs emphasize features that are approximately the same size as the fitting window, and fail to pick up fluctuations at different size scales. To perform a quantitative analysis which faithfully captures fluctuations at all size scales, we choose to perform Fourier decomposition on the leading edge shape, and analyze the dynamics of each wavemode separately. As cells migrate, their global

leading edge shape undergoes long timescale changes, such as variations in width, large-scale curvature, or slight turning of the cell, which can dominate the Fourier amplitudes and the subsequent autocorrelation signal. As we are most interested in extracting the fine-scale fluctuations, we performed background subtraction on the segmented leading edge shapes (*Figure 1—figure supplement 2*). To do this, we defined the 'background' leading edge shape as the contour after smoothing (by the lowess method, using a span of 7 μm). This rather large smoothing window was chosen specifically to preserve fine-scale features. The background-subtracted y-position is thus defined as the difference between the segmented leading edge and its smoothed counterpart. This process removed the large-scale features of the leading edge. We next wanted to remove the long-timescale features of the leading edge, so we also subtracted the time-averaged background-subtracted y-position for each x-pixel. Altogether, these pre-processing steps maintained the features of interest in the curvature kymograph (*Figure 1—figure supplement 3*). After performing background subtraction, we still needed to control for changes in leading edge width over time. The wavelengths represented in the Fourier transform are defined as $\lambda = L/n$, where $L$ is the length of the leading edge, and n is an integer from 0 to one half the number of pixels. If the leading edge length were to vary over time, then so would the wavelengths, making it impossible to track the behavior of a single wavelength fluctuation over time. We thus cropped the dataset along the x-axis to include only pixels which contain the cell for all timepoints in the video, thereby extracting a fixed-length leading edge subset for further analyses.

## Autocorrelation analysis and fitting

Autocorrelation analysis and fitting were performed separately for each cell and simulation. For experimental data, the entire video was analyzed. For simulated data, analysis was only performed on the time points after the simulation had reached steady state, for which we used a conservative cut-off of 10 s (see *Figure 2d–h*). To separate out fluctuations at different length scales, we first performed a spatial Fourier transform on the leading edge shape. Referencing the coordinate system defined in *Figure 2a*, the pixels (experiments) or membrane segments (simulations) are equally spaced in the x-direction, allowing us to perform a one-dimensional Fourier transform (MATLAB fft() function, which assumes periodic boundary conditions) on the y-positions of a segmented leading edge for each time point. We then normalize the Fourier transform by a factor of $dx/\sqrt{L}$, where $dx$ is the pixel/membrane segment size and $L$ is leading edge length. This normalization preserves the variance and accounts for the pixel size. To measure the fluctuation relaxation, we calculated the time-averaged autocorrelation ($A_n(\tau) = <Y_n(t+\tau)\cdot Y_n^*(t)>_t$, using non-overlapping windows in t) of each Fourier mode amplitude. The autocorrelation function extracted from this analysis contains complex elements of the form $A = a + ib$. We performed all plotting and fitting on the complex magnitude (sqrt ($a^2 + b^2$)) of the autocorrelation function, which is most representative of the total autocorrelation.

Note that because we are plotting the complex magnitude (which is always positive), the autocorrelation plots shown in *Figures 1e–f , and 2i–j* are expected to decay to some non-zero background noise window, rather than to zero. Indeed, the membrane simulation control (see *Validation of autocorrelation analysis implementation*), which is predicted to have a purely exponentially-decaying autocorrelation function, also shows a decay to a noise window at long times (*Figure 1—figure supplement 4*). We fit each Fourier mode time-autocorrelation to the exponential decay function described in the main text (*Figure 1f*), fitting $ln(|A_n(\tau)|)$ vs $\tau$ to a line using MATLAB's polyfit function, to extract fit parameters for each cell and simulation. Each curve was fit out to a drop in the amplitude by a factor of $e/2$, or at least 10 points. To average fit parameters over many cells, we controlled for cell-to-cell variability in leading edge length by binning the parameters by spatial frequency, and then calculating summary statistics separately for each bin.

The spatial background subtraction performed on the leading edge shape (discussed in Materials and methods: *Processing of segmented cell shapes for autocorrelation analysis*) was necessary to extract the fine-scale shape fluctuations studied in this work. This background subtraction is expected to remove fluctuations with wavelengths larger than ~7 μm (i.e. reduce their amplitude to zero). For this reason, only wavelengths less than 7 μm are plotted in *Figure 1e–h*. We note that it is possible that fluctuations with wavelengths less than, but near 7 μm might also have slightly reduced measured amplitudes (i.e. the shape of the curve in *Figure 1g* may artificially level off at low spatial frequency). However, any such effect would be performed uniformly in time, and therefore is not expected to

affect the measured temporal dynamics (*Figure 1h*). Indeed, when we extend the span of our background subtraction by ~50% (up to 10 µm), we find that only the amplitude of the largest mode is altered (slightly increased) and the measured relaxation timescales are not affected (*Figure 1—figure supplement 5*).

## Validation of autocorrelation analysis implementation

To validate our autocorrelation method, we analyzed control simulations of a membrane freely fluctuating under Brownian motion in the absence of actin, and showed it recapitulates predictions from analytical theory for this system (*Figure 1—figure supplement 3*). These simulations were performed exactly as in the leading edge simulations, using the same parameters, but without actin. The equation of motion used for membrane segments in the control simulations, as well as a derivation of the associated autocorrelation function, can be found in Appendix 2.4.4: Choice of timestep. Interestingly, the curvature kymographs of these control simulations exhibit striking visual features reminiscent of instabilities, dominant wavemodes, or oscillations – and yet such effects are absent from this system by definition (which we confirm quantitatively using our autocorrelation method, *Figure 1—figure supplement 3*). This suggests that similar features in the experimentally-measured curvature kymograph are also not indicative of instabilities, dominant wavemodes, or oscillations, which we confirmed by autocorrelation analysis.

## Modeling

Here, we briefly describe the geometry and major elements of the model. Please see the Appendix for a detailed description of the model and *Tables 1 and 2* for a list of the chosen parameters. A simulated patch of leading edge was modeled by a branched network of actin filaments stochastically polymerizing towards a 2D strip of membrane, subject to periodic boundary conditions. The 2D strip was discretized as membrane segments that are fixed in position along one axis and move only along the direction of motion of the simulated cell. Stochastic, fixed time step Brownian dynamics simulations, implemented with custom MATLAB code, were performed to update the membrane position and actin network properties. Actin network growth evolved from constant rate Poisson processes for polymerization, depolymerization, branching, and capping. Once polymerized, filaments were fixed in position at their branch point of origin (in the lab frame of reference), and did not undergo retrograde flow (i.e. translation of the filament position opposite the direction of migration) or translational diffusion. The membrane strip was subject to forces of bending and stretching, drag from fluid viscosity, as well as the force of actin (*Mogilner and Oster, 1996*; *Peskin et al., 1993*; *Carlsson, 2001*). Filaments apply force to the membrane segments according to the untethered Brownian ratchet formalism (*Mogilner and Oster, 1996*), in which filament pointed end positions are assumed to be rigidly connected to the network (via their branch point of origin) and their barbed end positions are able to freely fluctuate. As previously, we ignore the possibility of filament buckling due to the fact that lamellipodial filaments exist in a sufficiently low-load, high branch density regime (*Mogilner and Oster, 1996*), and experimental evidence shows no indication of buckling (*Svitkina et al., 1997*). We expanded this formalism (which previously only considered filament fluctuations perpendicular to the filament's long axis) to include all fluctuations of the filament along the filament's short and long axes. Each filament pushes the membrane segment that spans the growing tip's x-position. We note that the filament angles used to determine the filament forces on the membrane and presented throughout this work are always calculated relative to the global average direction of motion of the leading edge, rather than the local average membrane normal (a simplifying approximation necessitated by the discrete geometry and motivated by the shallow curvatures exhibited by cell leading edges). Control simulations were run to verify that the leading edge fluctuation behavior described in this work was not dependent on the temporal discretization (i.e. simulations were run with sufficiently small timesteps to resolve the fastest dynamics, *Figure 2—figure supplement 1*), spatial discretization (i.e. simulations were run with sufficiently short membrane segments to resolve the smallest length scales at which there is significant bending, *Figure 2—figure supplement 2*), or leading edge length (i.e. the periodic boundary conditions were implemented correctly, such that a simulated small patch of leading edge behaves identically to an equivalently sized portion of a larger simulated patch of leading edge, *Figure 2—figure supplement 3*). In cases where membrane tension and bending rigidity were

removed, these forces were simply not calculated in the simulation (*Figure 4a–e*). To remove filament spreading, we modified how the filament position was updated upon addition of a monomer in order to maintain the growing filament tip's x-position. Addition of monomers contributed only to changes in the barbed-end y-position, leaving the x-position intact, while updating the filament length correctly (effectively sliding the pointed end x-position backwards, rather than advancing the barbed end x-position forwards, *Figure 4f–i*).

## Acknowledgements

We dedicate this work to A B Savinov, who was in preparation alongside this manuscript and was born during revisions. We are exceedingly grateful to E Labuz for generously isolating and preparing fish epidermal keratocytes used for the experiments presented in *Figure 1—figure supplement 6*. We also kindly thank E F Koslover and A J Spakowitz for their thoughtful advice on model conceptualization and data analysis, and E F Koslover, A Mogilner, N M Belliveau, P Radhakrishnan, and A Savinov for helpful comments on the manuscript. Funding: Howard Hughes Medical Institute to JAT, Washington Research Foundation to JAT, Gerald J Lieberman Fellowship to RMG, NSF Graduate Research Fellowship to RMG.

## Additional information

### Funding

| Funder | Grant reference number | Author |
|---|---|---|
| National Science Foundation | | Rikki M Garner |
| Howard Hughes Medical Institute | | Julie A Theriot |
| Washington Research Foundation | | Julie A Theriot |

The funders had no role in study design, data collection and interpretation, or the decision to submit the work for publication.

### Author contributions

Rikki M Garner, Conceptualization, Data curation, Formal analysis, Investigation, Methodology, Software, Validation, Visualization, Writing – original draft, Writing – review and editing; Julie A Theriot, Conceptualization, Funding acquisition, Project administration, Resources, Supervision, Writing – review and editing

### Author ORCIDs

Rikki M Garner ⓘD http://orcid.org/0000-0001-9998-4596
Julie A Theriot ⓘD http://orcid.org/0000-0002-2334-2535

### Ethics

Experiments using zebrafish larvae were approved by the University of Washington Institutional Animal Care and Use Committee (protocol 4427-01).

### Decision letter and Author response

Decision letter https://doi.org/10.7554/eLife.74389.sa1
Author response https://doi.org/10.7554/eLife.74389.sa2

## Additional files

### Supplementary files
• Transparent reporting form

## Data availability

Analysis and modeling code for this paper is available on the Theriot lab Gitlab: <https://gitlab.com/theriot_lab/leading-edge-stability-in-motile-cells-is-an-emergent-property-of-branched-actin-network-growth> under the MIT license. Figure data are available in the Source Data files. The large size of the raw video microscopy data (865 GB of image files in the Open Microscopy Environment OME-TIFF format) and the associated analyzed data (320 GB) prohibits their upload to a public repository. The complete raw and analyzed data files for one example experimental dataset and one example simulated dataset (corresponding to the data shown in Fig. 1a-f and Fig. 2c-j, respectively) are available on Figshare <https://figshare.com/projects/Leading_edge_stability_in_motile_cells_is_an_emergent_property_of_branched_actin_network_growth/132878>. Code to analyze this data are publicly available on Gitlab as noted above. Requests for additional raw or analyzed data should be sent to the corresponding author by email. Data will be made available in the form of a hard drive shipped by mail. There are no restrictions on who may access the data.

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

## Appendix 1

### Detailed description of the model

In this section, we outline the main features of the model and reference the Detailed Derivations for more detailed derivations.

### Model geometry

Given the flat structure of lamellipodial protrusions, we considered leading edge dynamics in two dimensions. In cartesian coordinates, the cell migrates in the x-y plane along the y-axis and maintains a fixed leading edge height of 200 nm along the z-axis. The leading edge membrane was modeled as a 2D strip that restricts bending and stretching in the x-y plane and is perfectly flat along the z-axis. We discretized the membrane such that a 20 µm leading edge membrane was modeled as 200 flat, rectangular segments (each 100 nm in length along the x-axis and 200 nm in height along z-axis) whose surface normal is fixed to lie along the y-axis. The membrane is implented using a Monge parameterization, such that these membrane segments are fixed in position along the x- and z-axes and move only along the y-axis (the direction of motion of the simulated cell).

### Updating the membrane position

The membrane segments are assumed to move in a viscous medium at low Reynolds number, with drag force $F_{drag} = -\gamma \frac{\partial y}{\partial t}$ and Stoke's drag coefficient $\gamma = 6\pi\eta r$, where r is the membrane segment length (x-axis) and $\eta$ is the dynamic viscosity of water. The membrane also acts under the forces of membrane bending and stretching ($F_{S/B}$, see Appendix section 1.3), and Brownian ratchet forces by the actin filaments ($F_{BR}$, see Appendix section 1.4). This gives us the following equation of motion.

$$\sum F = 0 = -\gamma \frac{\partial y}{\partial t} + F_{S/B} + \sum_{n=1}^{N_{fil}} F_{actin}^n \qquad (1)$$

Brownian dynamics simulations, implemented with a 4th order Runge-Kutta algorithm, were performed to update the membrane segment positions. Thermal fluctuations of the membrane were ignored in this implementation, as thermal fluctuations of the (much stiffer) actin filaments dominate the membrane's motion as well as monomer incorporation into the actin network (*Mogilner and Oster, 1996*). See Appendix Detailed Derivations IV for a discussion of the numerical approximations included in the simulations.

### Forces of membrane bending and stretching

The simulated leading edge membrane acts under the energetic constrains of stretching and bending, characterized by the experimentally measurable parameters of membrane tension ($\sigma$, $pN \cdot nm^{-1}$) and bending modulus ($\kappa$, $pN \cdot nm$), and using the following energy functional:

$$\begin{aligned} E_{S/B} &= \int \int \frac{\sigma}{2} \left(\frac{\partial y}{\partial x}\right)^2 + \frac{\kappa}{2} \left(\frac{\partial^2 y}{\partial x^2}\right)^2 dx\, dz \\ &= h \int \frac{\sigma}{2} \left(\frac{\partial y}{\partial x}\right)^2 + \frac{\kappa}{2} \left(\frac{\partial^2 y}{\partial x^2}\right)^2 dx \end{aligned} \qquad (2)$$

where $h$ is the height of the leading edge in the z-dimension, $\frac{\partial y}{\partial x}$ is extension of the membrane (i.e. an increase in contour length), and $\frac{\partial^2 y}{\partial x^2}$ is curvature in (bending of) the membrane. The force of membrane stretching and bending on a single membrane segment ($F_{S/B}$) is defined as the free energy gained by movement of the segment (*seg*), and takes the following form... (See Detailed Derivations I for a derivation of the functional derivative and resulting force.)

$$\begin{aligned} F_{S/B} &= - \int_s eg \frac{\delta E_{S/B}}{\delta y} dx \\ &= \sigma_{\text{eff}} \frac{\partial^2 y}{\partial x^2} - \kappa_{\text{eff}} \frac{\partial^4 y}{\partial x^4} \end{aligned} \qquad (3)$$

### Actin filament Brownian ratchet forces

Actin filaments constantly undulate due to thermal fluctuations, bending and stretching to sample their conformational space. The presence of the membrane restricts fluctuations of the filament past the membrane, presenting an entropic cost and a reduction in the free energy of the filament. The

force of the filament exerted on the membrane segment, $F_{actin}$, can therefore be calculated as the gain in free energy, G, by an infinitesmal movement of the membrane position, $\xi$:

$$G = -k_B T \log Z_{conf} \tag{4}$$

$$F_{actin} = \left. \frac{\partial G}{\partial \xi} \right|_{\xi=0} \tag{5}$$

The partition function, $Z_{conf}$, determined by the energetic cost of bending the filament, $E_{bend}$, defines the conformational landscape. A few considerations must be made to determine the partition funciton in our system. Each filament applies force only to the membrane segment under which the filament's barbed (growing) end equilibrium position sits, making the approximation that each membrane segment acts as an infinite wall past which filament fluctuations are blocked. Given this assumption, the membrane only restricts filament fluctuations along the y-axis. The partition function is therefore integrated over all x-positions, but only the subset of y-positions where the filament is not restricted by the membrane. With these considerations in mind, we arrive at our partition function: (See Appendix Detailed Derivations II for a derivation of the filament bending energy and Detailed Derivations III for the derivation of the force).

$$Z_{conf} = \int_{-\infty}^{\infty} \int_{0}^{\infty} e^{-\beta E_{bend}} \, dy \, dx \tag{6}$$

where y is measured relative to the membrane surface and $\beta = \frac{1}{k_B T}$. The final equation for the force of a filament on the membrane becomes…

$$F_{actin} = \frac{k_B T e^{-\frac{1}{2} \bar{\kappa}_{Eff} y_0^2}}{\int_{0}^{\infty} e^{-\frac{1}{2} \bar{\kappa}_{Eff} (y - y_0)^2} \, dy} \tag{7}$$

where $y_0$ is the equilibrium filament position measured relative to the membrane surface (the filament pokes through the membrane for $y_0 < 0$) and

$$\bar{\kappa}_{Eff} = \left( \frac{\bar{\kappa}_{\parallel} \bar{\kappa}_{\perp}}{\bar{\kappa}_{\perp} cos^2(\theta) + \bar{\kappa}_{\parallel} sin^2(\theta)} \right) \tag{8}$$

and $\bar{\kappa}_{\perp} = \frac{3}{L^2 N}$, $\bar{\kappa}_{\parallel} = \frac{16}{L^2 N^2}$, and $N = \frac{L}{l_p}$ for a filament with length L and persistence length $l_p$ at an angle $\theta$ relative to the membrane segment normal.

## Actin network dynamics

Actin network dynamics including polymerization, depolymerization, branching, and capping were assumed to be independent, constant rate Poisson processes. The choice of rates are described in their respective sections below. For a given time step of $\Delta t$ and rate $r$, the probability of an event happening during any given time step is…

$$p = 1 - e^{-r \cdot \Delta t} \tag{9}$$

where $e^{-r \cdot \Delta t}$ is the probability that the event did NOT take place within a time step of $\Delta t$. A random number generator was used to determine which processes occurred in each time step. In particular, a random number was chosen between 0 and 1. If the random number lied below the probability $p$, then the event occured. For each simulation the random number generator was seeded with a unique, semi-random number, based on the current time.

### Polymerization and depolymerization

The rate of polymerization was assumed to be $r_{on} = k_{on} \cdot M \cdot gap'$, where $k_{on}$ is the rate of polymerization per free monomer concentration in monomers $ms^{-1} \mu M^{-1}$, M is the free monomer concentration in $\mu M$, and $p_{gap}$ is the probability that enough space opens up in between the filament tip and the membrane to add a monomer (such that the polymerization rate far away from the membrane is $r_{(y=\infty)} = k_{on} \cdot M$). Previously, it was determined that this probability is set by thermal fluctuations of the filament (*Mogilner and Oster, 1996*). In other words, the probability of adding a monomer is determined by the probability that thermal fluctuations, by chance, overcome the bending and

stretching energies of the filament and bend the filament tip away from the membrane enough to open up a space of sufficient size to add a monomer. Given a monomer width $\Delta$, the probability of adding a monomer is…

$$p_{gap} = \frac{\int_{\Delta}^{\infty} e^{-\frac{1}{2}\kappa_{\mathrm{Eff}}(y-y_0)^2} dy}{\int_{0}^{\infty} e^{-\frac{1}{2}\kappa_{\mathrm{Eff}}(y-y_0)^2} dy} \tag{10}$$

where $\kappa_{\mathrm{Eff}}$ takes into account the thermal energy ($k_B T$) as well as the flexibility, length, and orientation of the filament. Depolymerization was assumed to be a constant rate process with rate $r_{off}$ (monomers $ms^{-1}$), independent of membrane proximity.

### Branching

The rate of branching was calculated for each filament at each time step, such that $r_{\mathrm{branch}} = k_{\mathrm{branch}} \cdot M \cdot l_{\mathrm{branch}}$. Here $k_{\mathrm{branch}}$ is the branching rate per free monomer concentration per length of mother filament in units of branches $ms^{-1}\mu M^{-1}nm^{-1}$, M is the free monomer concentration in $\mu M$, and $l_{\mathrm{branch}}$ is the length of the filament that sits inside the branching window. A fixed-length branching window is required for model stability, and is well-supported by experimental evidence that branching activation is localized to the leading edge membrane. Unlike linear polymerization, branching was not inhibited by membrane proximity. For simplicity, new branches were placed on the tip of the mother filament. If this placement caused the new branch tip position to extend past the membrane, then the branches were placed on the side of the mother filament such that the branch tip is flush with the membrane. The angle of the branch relative to the mother filament was randomly selected from a normal distribution with mean $\mu = \theta_{branch}$ and standard deviation $\sigma = \Delta\theta_{branch}$ as specified for each simulation. The side of the mother filament on which the branch was placed was random.

### Capping

Capping was assumed to be a constant rate process with rate $r_c$ ($ms^{-1}$), independent of membrane proximity. Filaments were not allowed to uncap. Capped filaments were not allowed to polymerize or depolymerize.

### Filament deletion

Our simulations were intended to capture only leading edge actin dynamics. We therefore chose, for simulation efficiency, to only keep track of filaments actively applying force to the membrane. Filaments which were both (1) capped and (2) cumulatively provided less than 0.1% of the force on a given membrane segment were deleted from the simulation.

## Detailed Derivations

This Detailed Derivations contains detailed derivations and clarifications for the material discussed in the previous section of the Appendix: "Detailed description of the model".

## Detailed Derivations I: Forces of membrane bending and stretching

In this section, we use the energy functional for membrane elasticity defined in section A.3 to calculate the elastic forces on a discrete membrane segment, filling in the steps of Appendix *equation (3)*.

### Functional derivative of the membrane stretch/bend energy functional

We can solve the functional derivative $\frac{\delta E}{\delta y}$ for our particular energy function (Appendix *equation (2)*) using the fact that for a functional of the following type…

$$F[\rho] = \int f(r, \rho(r), \nabla\rho(r), ..., \nabla^N \rho) dr$$

The functional derivative is calculated by….

$$\frac{\delta F}{\delta \rho(r)} = \left( \frac{\partial f}{\partial \rho} + \sum_{i=1}^{N} (-1)^i \nabla^i \cdot \frac{\partial f}{\partial \nabla^i \rho} \right)$$

So for this particular energy functional

$$E[y] = h \int \frac{\sigma}{2} \left( \frac{\partial y}{\partial x} \right)^2 + \frac{\kappa}{2} \left( \frac{\partial^2 y}{\partial x^2} \right)^2 dx \tag{11}$$

$$f(x, y(x), \nabla y(x), ..., \nabla^N y(x)) = h \left( \frac{\sigma}{2} \left( \frac{\partial y}{\partial x} \right)^2 + \frac{\kappa}{2} \left( \frac{\partial^2 y}{\partial x^2} \right)^2 \right) \tag{12}$$

$$
\begin{aligned}
\frac{\delta E}{\delta y(x)} &= h \left( -\frac{d}{dx} \frac{df}{d\left( \frac{\partial y}{\partial x} \right)} + \frac{d^2}{dx^2} \frac{df}{d\left( \frac{\partial^2 y}{\partial x^2} \right)} \right) \\
&= h \left( -\frac{d}{dx} \left( \sigma \left( \frac{\partial y}{\partial x} \right) \right) + \frac{d^2}{dx^2} \left( \kappa \left( \frac{\partial^2 y}{\partial x^2} \right) \right) \right) \\
&= h \left( -\sigma \frac{\partial^2 y}{\partial x^2} + \kappa \frac{\partial^4 y}{\partial x^4} \right)
\end{aligned}
\tag{13}
$$

Note that $\frac{\delta E}{\delta y(x)}$ is a functional derivative (rather than an ordinary derivative), with units of a force per unit length (rather than a force). Finally, we can calculate the total elastic force on a discrete membrane segment of length $\Delta x$, by integrating this force density over the length of a segment (within which the spatial derivatives of $y$ do not vary).

$$
\begin{aligned}
F_{S/B} &= -\int_{seg} \frac{\delta E_{S/B}}{\delta y} dx \\
&= -\int_{seg} h \left( -\sigma \frac{\partial^2 y}{\partial x^2} + \kappa \frac{\partial^4 y}{\partial x^4} \right) dx \\
&= h \Delta x \left( \sigma \frac{\partial^2 y}{\partial x^2} - \kappa \frac{\partial^4 y}{\partial x^4} \right) \\
&= \sigma_{\text{eff}} \frac{\partial^2 y}{\partial x^2} - \kappa_{\text{eff}} \frac{\partial^4 y}{\partial x^4}
\end{aligned}
\tag{14}
$$

where $\sigma_{\text{eff}} = \sigma h \Delta x$ and $\kappa_{\text{eff}} = \kappa h \Delta x$.

## Detailed Derivations II: 2D thermal fluctuations of actin filaments

In this section, we characterize the thermal fluctuations of an actin filament with a given length and persistence length (and the associated bending modulus). We first decompose the fluctuations into their (small) end-to-end and (larger) side-to-side fluctuations, and then further perform a wavelength decomposition on the side-to-side fluctuations. We next apply the equipartition theorem, giving $\frac{1}{2} k_B T$ to each independent bending mode, to determine the fluctuation magnitude of each of these modes. We then determine the average total side-to-side and end-to-end fluctuation magnitudes, allowing us to calculate effective bending coefficients for the side-to-side ($\kappa_\perp$) and end-to-end ($\kappa_\parallel$) fluctuations. These effective bending energies will then be used in Detailed Derivations III to determine the force of a filament on a membrane segment.

### Determining the wavelength-dependence of thermal fluctuations

Assume we have a filament of length $L$, which, in the absence of thermal fluctuations, points vertically upward in the direction $\hat{i}$. Due to thermal fluctuations, the actin polymer will fluctuate all along its length, as well as along the directions of both the short and long axis of the filament. At any point $s$ along the polymer, we can define the local position vector $\vec{r}(s)$, and the local tangent vector of the polymer $\vec{t}(s) = \frac{\partial \vec{r}}{\partial s}$. Given a persistence length $l_p$, and thus a bending modulus $B = l_p k_B T$, the bending energy $E_{bend}$ of a filament is defined as…

$$
\begin{aligned}
E_{bend} &= \frac{B}{2} \int_0^L \left( \frac{\partial \vec{t}}{\partial s} \right)^2 ds \\
\beta E_{bend} &= \frac{l_p}{2} \int_0^L \left( \frac{\partial \vec{t}}{\partial s} \right)^2 ds
\end{aligned}
\tag{15}
$$

where $\beta = \frac{1}{k_B T}$ and $\vec{t}(s = 0) = \hat{i}$. If we assume the thermal fluctuations are small ($\frac{L}{l_p} << 1$), then we can write the position vector along the polymer as…

$$\vec{r}(s) = J(s)\hat{j} + \left( s - \delta(s) \right) \hat{i} \tag{16}$$

Then…

$$\vec{t} = \frac{\partial \vec{r}}{\partial s} = \frac{\partial J}{\partial s}\hat{j} + \left(1 - \frac{\partial \delta}{\partial s}\right)\hat{i} \tag{17}$$

And...

$$\frac{\partial \vec{t}}{\partial s} = \frac{\partial^2 J}{\partial s^2}\hat{j} - \frac{\partial^2 \delta}{\partial s^2}\hat{i} \tag{18}$$

Because $\vec{t}$ is the unit tangent vector, we know $\vec{t} \cdot \vec{t} = 1$. This allows us to solve for $\frac{\partial \delta}{\partial s}$ in terms of $\frac{\partial J}{\partial s}$.

$$
\begin{aligned}
\vec{t} \cdot \vec{t} = 1 &= \left(\frac{\partial J}{\partial s}\right)^2 + \left(1 - \frac{\partial \delta}{\partial s}\right)^2 \\
0 &\approx \left(\frac{\partial J}{\partial s}\right)^2 - 2\left(\frac{\partial \delta}{\partial s}\right) \\
\frac{\partial \delta}{\partial s} &\approx \frac{1}{2}\left(\frac{\partial J}{\partial s}\right)^2 \\
\frac{\partial^2 \delta}{\partial s^2} &\approx \frac{\partial J}{\partial s}\frac{\partial^2 J}{\partial s^2}
\end{aligned} \tag{19}
$$

Plugging this back into our equation for $E_{bend}$...

$$
\begin{aligned}
\beta E_{bend} &= \frac{l_p}{2}\int_0^L \left(\frac{\partial \vec{t}}{\partial s}\right)^2 ds \\
&= \frac{l_p}{2}\int_0^L \left(\frac{\partial \vec{t}}{\partial s} \cdot \frac{\partial \vec{t}}{\partial s}\right) ds \\
&= \frac{l_p}{2}\int_0^L \left(\left(\frac{\partial^2 J}{\partial s^2}\right)^2 + \left(\frac{\partial^2 \delta}{\partial s^2}\right)^2\right) ds \\
&= \frac{l_p}{2}\int_0^L \left(\left(\frac{\partial^2 J}{\partial s^2}\right)^2 + \left(\frac{\partial J}{\partial s}\frac{\partial^2 J}{\partial s^2}\right)^2\right) ds \\
&\approx \frac{l_p}{2}\int_0^L \left(\frac{\partial^2 J}{\partial s^2}\right)^2 ds
\end{aligned} \tag{20}
$$

By decomposing $J(s)$ into its wavemodes, we can determine the relative amplitudes $A_n$ of thermal fluctuations at different lengthscales, where $n$ refers to the specific wavemode. This choice of wavemode decomposition oscillates around $\sqrt{2}A_n$ from 0 to $2\sqrt{2}A_N$ ($\langle A_n \rangle = 0$), where the filament is pinned at zero at $s = 0$ and open at the other end.

$$
\begin{aligned}
J(s) &= \sum_{n=0}^{\infty} \sqrt{2}A_n \left[1 - cos\left(\pi\left(n+\frac{1}{2}\right)\frac{s}{L}\right)\right] \\
\frac{\partial J}{\partial s} &= \sum_{n=0}^{\infty} \sqrt{2}A_n\pi\left(n+\frac{1}{2}\right)\frac{1}{L}sin\left(\pi\left(n+\frac{1}{2}\right)\frac{s}{L}\right) \\
\frac{\partial^2 J}{\partial s^2} &= \sum_{n=0}^{\infty} \sqrt{2}A_n\pi^2\left(n+\frac{1}{2}\right)^2\frac{1}{L^2}cos\left(\pi\left(n+\frac{1}{2}\right)\frac{s}{L}\right) \\
\left(\frac{\partial^2 J}{\partial s^2}\right)^2 &= \sum_{n=0}^{\infty} 2A_n^2\pi^4\left(n+\frac{1}{2}\right)^4\frac{1}{L^4}cos^2\left(\pi\left(n+\frac{1}{2}\right)\frac{s}{L}\right) \\
&+ \sum_{n\neq m} 2A_n \cdot A_m\pi^4\left(n+\frac{1}{2}\right)^2\left(m+\frac{1}{2}\right)^2\frac{1}{L^4} \quad cos\left(\pi\left(n+\frac{1}{2}\right)\frac{s}{L}\right) \\
&\quad cos\left(\pi\left(m+\frac{1}{2}\right)\frac{s}{L}\right)
\end{aligned} \tag{21}
$$

Because...

$$\left(\sum_{n=0}^{N} A_n\right)^2 = \sum_{n=0}^{N} A_n^2 + \sum_{n\neq m} A_n \cdot A_m \tag{22}$$

Now we can solve the integral for the parts of the function that contain $L$. For the part where $n = m$...

$$
\begin{aligned}
\int_0^L &cos^2\left(\pi\left(n+\frac{1}{2}\right)\frac{s}{L}\right) ds \\
&= \frac{2\left(\pi\left(n+\frac{1}{2}\right)\right)+sin\left(2\pi\left(n+\frac{1}{2}\right)\right)}{4\left(\frac{\pi}{L}\left(n+\frac{1}{2}\right)\right)} \\
&= \frac{L}{2}
\end{aligned} \tag{23}
$$

And for $n \neq m$...

$$\int_0^L cos\left(\pi\left(n+\frac{1}{2}\right)\frac{s}{L}\right) cos\left(\pi\left(m+\frac{1}{2}\right)\frac{s}{L}\right) ds \tag{24}$$

$$
\begin{aligned}
\beta E_{bend} &\approx \frac{l_p}{2}\int_0^L \left(\frac{\partial^2 J}{\partial s^2}\right)^2 ds \\
&= \frac{l_p}{2}\sum_{n=0}^{\infty} 2A_n^2\pi^4\left(n+\frac{1}{2}\right)^4\frac{1}{L^4}\frac{L}{2} \\
&= \frac{l_p\pi^4}{2L^3}\sum_{n=0}^{\infty} A_n^2\left(n+\frac{1}{2}\right)^4 \\
&= \frac{\pi^4}{2L^2 N}\sum_{n=0}^{\infty} A_n^2\left(n+\frac{1}{2}\right)^4
\end{aligned}
\tag{25}
$$

where $N = \frac{L}{l_p}$. By the equipartition theorem ($\beta E_{bend} = 1/2$ for 1 degree of freedom), we can determine the Fourier coefficients.

$$\left\langle A_n^2\right\rangle = \frac{L^2 N}{\pi^4\left(n+\frac{1}{2}\right)^4} \tag{26}$$

We can now calculate a few useful integrals. From the equipartition theorem and Appendix *Equation 26*:

$$\left\langle\int_0^L\left(\frac{\partial^2 J}{\partial s^2}\right)^2 ds\right\rangle = \frac{N}{L} \tag{27}$$

$$
\begin{aligned}
\left\langle\int_0^L\left(\frac{\partial J}{\partial s}\right)^2 ds\right\rangle &= \sum_{n=0}^{\infty} 2\left\langle A_n^2\right\rangle\pi^2\left(n+\frac{1}{2}\right)^2\frac{1}{L^2}\frac{L}{2} \\
&= \frac{1}{L}\sum_{n=0}^{\infty}\frac{L^2 N}{\pi^4\left(n+\frac{1}{2}\right)^4}\pi^2\left(n+\frac{1}{2}\right)^2 \\
&= \frac{LN}{\pi^2}\sum_{n=0}^{\infty}\frac{1}{\left(n+\frac{1}{2}\right)^2} \\
&= \frac{LN}{2}
\end{aligned}
\tag{28}
$$

Because…

$$\sum_{n=0}^{\infty}\frac{1}{\left(n+\frac{1}{2}\right)^2} = \frac{\pi^2}{2} \tag{29}$$

Finally…

$$
\begin{aligned}
\left\langle\left(\int_0^L\left(\frac{\partial J}{\partial s}\right)^2 ds\right)^2\right\rangle &= \left\langle\left(\sum_{n=0}^{\infty} 2A_n^2\pi^2\left(n+\frac{1}{2}\right)^2\frac{1}{L^2}\frac{L}{2}\right)^2\right\rangle \\
&= \sum_{n=0}^{\infty} 4\left\langle A_n^4\right\rangle\pi^4\left(n+\frac{1}{2}\right)^4\frac{1}{L^4}\frac{L^2}{4} \\
&= \sum_{n=0}^{\infty} 4\frac{3L^4 N^2}{\pi^8\left(n+\frac{1}{2}\right)^8}\pi^4\left(n+\frac{1}{2}\right)^4\frac{1}{L^4}\frac{L^2}{4} \\
&= \frac{3L^2 N^2}{\pi^4}\sum_{n=0}^{\infty}\frac{1}{\left(n+\frac{1}{2}\right)^4} \\
&= \frac{3L^2 N^2}{\pi^4}\frac{\pi^4}{6} \\
&= \frac{L^2 N^2}{2}
\end{aligned}
\tag{30}
$$

Because for a Gaussian distribution, the 4th moment is related to the variance in the following way…

$$\left\langle A_n^4\right\rangle = 3\left\langle A_n^2\right\rangle^2 = \frac{3L^4 N^2}{\pi^8\left(n+\frac{1}{2}\right)^8} \tag{31}$$

and

$$\sum_{n=0}^{\infty} \frac{1}{\left(n+\frac{1}{2}\right)^4} = \frac{\pi^4}{6} \tag{32}$$

## Determining the average end-to-end retraction

The the new effective length of the filament is $I(s=L)$ can be found by integrating $\hat{i}$ component of the tangent vector $\vec{t}$ over the arc length of the filament $s$.

$$
\begin{aligned}
I_{s=L} &= \hat{i} \cdot \int_0^L \vec{t}\, ds \\
&= \int_0^L \left(1 - \frac{\partial \delta}{\partial s}\right) ds \\
&\approx \int_0^L \left(1 - \frac{1}{2}\left(\frac{\partial J}{\partial s}\right)^2\right) ds \\
&\approx L - \frac{1}{2}\int_0^L \left(\frac{\partial J}{\partial s}\right)^2 ds
\end{aligned}
\tag{33}
$$

From Appendix *Equation 28*...

$$
\begin{aligned}
\langle I_{s=L} \rangle &\approx L - \frac{1}{2}\frac{LN}{2} \\
&\approx L - \frac{LN}{4} \\
&\approx L\left(1 - \frac{N}{4}\right)
\end{aligned}
\tag{34}
$$

Determining the end-to-end retraction fluctuations

$$
\begin{aligned}
\left(I_{s=L} - \langle I_{s=L}\rangle\right)^2 &= \left(L - \frac{1}{2}\int_0^L \left(\frac{\partial J}{\partial s}\right)^2 ds - L\left(1 - \frac{N}{4}\right)\right)^2 \\
&= \left(-\frac{1}{2}\int_0^L \left(\frac{\partial J}{\partial s}\right)^2 ds + \frac{LN}{4}\right)^2 \\
&= \frac{L^2 N^2}{16} - \frac{LN}{4}\int_0^L \left(\frac{\partial J}{\partial s}\right)^2 ds + \frac{1}{4}\left(\int_0^L \left(\frac{\partial J}{\partial s}\right)^2 ds\right)^2
\end{aligned}
\tag{35}
$$

From Appendix *Equation 28* and *Equation 30*...

$$
\begin{aligned}
\left\langle \left(I_{s=L} - \langle I_{s=L}\rangle\right)^2 \right\rangle &= \frac{L^2 N^2}{16} - \frac{LN}{4}\frac{LN}{2} + \frac{1}{4}\frac{L^2 N^2}{2} \\
&= \frac{L^2 N^2}{16}
\end{aligned}
\tag{36}
$$

Determining the side-to-side fluctuations

$$
\begin{aligned}
J_{s=L} &= \hat{j} \cdot \int_0^L \vec{t}\, ds \\
&= \int_0^L \frac{\partial J}{\partial s} ds \\
&= \int_0^L \sum_{n=0}^{\infty} \sqrt{2}A_n \pi \left(n+\frac{1}{2}\right)\frac{1}{L}\sin\left(\pi\left(n+\frac{1}{2}\right)\frac{s}{L}\right) ds \\
&= \sum_{n=0}^{\infty} \sqrt{2}A_n
\end{aligned}
\tag{37}
$$

$$
\begin{aligned}
\left\langle J_{s=L}^2 \right\rangle &= \sum_{n=0}^{\infty} 2\left\langle A_n^2 \right\rangle \\
&= \sum_{n=0}^{\infty} 2\frac{L^2 N}{\pi^4 \left(n+\frac{1}{2}\right)^4} \\
&= \frac{2L^2 N}{\pi^4} \sum_{n=0}^{\infty} \frac{1}{\left(n+\frac{1}{2}\right)^4} \\
&= \frac{2L^2 N}{\pi^4}\frac{\pi^4}{6} \\
&= \frac{L^2 N}{3}
\end{aligned}
\tag{38}
$$

## Deriving effective stretching constants

We know by the equipartition theorem that for each degree of freedom, $\frac{1}{2}k_BT = \frac{1}{2}\kappa\left\langle x^2\right\rangle$. So for the $\hat{x}$ direction...

$$
\begin{aligned}
\frac{1}{2}\kappa_\perp\left\langle J_{s=L}^2\right\rangle &= \frac{1}{2}k_BT \\
\frac{1}{2}\kappa_\perp\frac{L^2N}{3} &= \frac{1}{2}k_BT \\
\kappa_\perp &= k_BT\frac{3}{L^2N}
\end{aligned}
\tag{39}
$$

And for the $\hat{i}$ direction...

$$
\begin{aligned}
\frac{1}{2}\kappa_\parallel\left\langle\left(I_{s=L}-\left\langle I_{s=L}\right\rangle\right)^2\right\rangle &= \frac{1}{2}k_BT \\
\frac{1}{2}\kappa_\parallel\frac{L^2N^2}{16} &= \frac{1}{2}k_BT \\
\kappa_\parallel &= k_BT\frac{16}{L^2N^2}
\end{aligned}
\tag{40}
$$

This gives us an effective bending energy as a function of actin filament tip position....

$$
\beta E_{bend} = \frac{1}{2}\bar{\kappa}_\perp J_{s=L}^2 + \frac{1}{2}\bar{\kappa}_\parallel\left(I_{s=L}-L+\frac{NL}{4}\right)^2
\tag{41}
$$

where $\bar{\kappa}_\perp = \frac{3}{L^2N}$ and $\bar{\kappa}_\parallel = \frac{16}{L^2N^2}$.

## Detailed Derivations III: The force of an actin filament on the membrane

In this section, we take the effective side-to-side and end-to-end filament bending energies calculated in Detailed Derivations II to determine the force of a filament on a membrane segment. We start by converting from the coordinate system used in Detailed Derivations II (in the frame of the filament long axis) to the reference frame of the membrane segment. We then use the bending energies to evaluate the partition function $Z_{conf}$ for a filament which is constrained by the membrane segment - and then use the partition function to determine the free energy $G$ of the filament. Finally, we evaluate the force of the filament on the membrane segment ($F_{actin}$) as the increase in filament free energy obtained by an incremental movement of the membrane $\left.\frac{\partial G}{\partial\xi}\right|_{\xi=0}$.

### Converting to the reference frame of the membrane segment

We derived this bending energy function in the reference frame of the filament. However, our partition function, $Z_{conf}$, will need to be integrated across the x-y coordinate system used in the rest of the paper. The energy function can be extended to an arbitrary filament orientation in a $\vec{u}_0$ and tip position $\vec{r}$ relative to resting filament tip position $\vec{r}_0 = (0,0)$ in the following way:

$$
\beta E_{bend} = \frac{1}{2}\bar{\kappa}_\perp\left(\vec{r}\left(\underline{I}-\left(\vec{u}_0\otimes\vec{u}_0\right)\right)\vec{r}\right) + \frac{1}{2}\bar{\kappa}_\parallel\left(\vec{r}\cdot\vec{u}_0\right)^2
\tag{42}
$$

Taking a coordinate system centered around the equilibrium filament tip position and aligned with the membrane normal, a filament lying at angle $\theta$ relative to the membrane normal will have filament orientation vector $\vec{u}_0 = sin(\theta)\hat{x} + cos(\theta)\hat{y}$ and filament tip position $\vec{r} = x\hat{x} + y\hat{y}$ will having the following perpendicular and parallel displacements...

$$
\begin{aligned}
\vec{r}\cdot\vec{u}_0 &= xsin(\theta)+ycos(\theta) \\
\left(\vec{r}\cdot\vec{u}_0\right)^2 &= x^2sin^2(\theta)+y^2cos^2(\theta)+2xycos(\theta)sin(\theta)
\end{aligned}
\tag{43}
$$

and...

$$
\begin{aligned}
\vec{r}\left(\underline{I}-\left(\vec{u}_0\otimes\vec{u}_0\right)\right)\vec{r} &= (\vec{r}\cdot\vec{r})^2-(\vec{r}\cdot\vec{u}_0)^2 \\
&= x^2+y^2-y^2cos^2(\theta)-x^2sin^2(\theta)-2xycos(\theta)sin(\theta) \\
&= x^2cos^2(\theta)+y^2sin^2(\theta)-2xycos(\theta)sin(\theta)
\end{aligned}
\tag{44}
$$

We arrive at the following bending energy as a function of the filament angle, relative to the membrane normal.

$$
\begin{aligned}
\beta E_{\text{bend}} &= \tfrac{1}{2}\bar{\kappa}_\perp \left(x^2 cos^2(\theta) + y^2 sin^2(\theta) - 2xy cos(\theta)sin(\theta)\right) \\
&+ \tfrac{1}{2}\bar{\kappa}_\parallel \left(x^2 sin^2(\theta) + y^2 cos^2(\theta) + 2xy cos(\theta)sin(\theta)\right) \\
&= \left(\tfrac{\bar{\kappa}_\perp}{2}cos^2(\theta) + \tfrac{\bar{\kappa}_\parallel}{2}sin^2(\theta)\right)x^2 + \left(\tfrac{\bar{\kappa}_\perp}{2}sin^2(\theta) + \tfrac{\bar{\kappa}_\parallel}{2}cos^2(\theta)\right)y^2 \\
&+ \left(\left(\tfrac{\bar{\kappa}_\parallel}{2} - \tfrac{\bar{\kappa}_\perp}{2}\right)2sin(\theta)cos(\theta)\right)xy
\end{aligned}
\tag{45}
$$

The x- and y-components have been separated for ease of integration in the next section.

## Simplification of the bending energy function

Currently, the exponential being integrated is of the form $\int_{-\infty}^{\infty}\int_0^{\infty} e^{-ax^2+bxy-cy^2} dy dx$. We can re-write the integral in the form $\int_{-\infty}^{\infty}\int_0^{\infty} e^{-(dx+fy)^2 - gy^2} dy dx$, to take advantage of the fact that $\int_{-\infty}^{\infty} e^{-(x+c)^2}dx = \int_{-\infty}^{\infty} e^{-x^2}dx$. To do this, we can complete the square: $f(x) = ax^2 + bx + c = a\left(x + \tfrac{b}{2a}\right)^2 + c - \tfrac{b^2}{4a}$

$$
\begin{aligned}
\beta E_{\text{bend}} &= \left(\tfrac{\kappa_\perp}{2}cos^2(\theta) + \tfrac{\kappa_\parallel}{2}sin^2(\theta)\right)\left(x + \frac{\left(\left(\tfrac{\kappa_\parallel}{2} - \tfrac{\kappa_\perp}{2}\right)2y sin(\theta)cos(\theta)\right)}{2\left(\tfrac{\kappa_\perp}{2}cos^2(\theta) + \tfrac{\kappa_\parallel}{2}sin^2(\theta)\right)}\right)^2 \\
&+ \left(\tfrac{\kappa_\perp}{2}sin^2(\theta) + \tfrac{\kappa_{parallel}}{2}cos^2(\theta)\right)y^2 - \frac{\left(\left(\tfrac{\kappa_\parallel}{2} - \tfrac{\kappa_\perp}{2}\right)2y sin(\theta)cos(\theta)\right)^2}{4\left(\tfrac{\kappa_\perp}{2}cos^2(\theta) + \tfrac{\kappa_\parallel}{2}sin^2(\theta)\right)} \\
&= \left(\tfrac{\kappa_\perp}{2}cos^2(\theta) + \tfrac{\kappa_\parallel}{2}sin^2(\theta)\right)\left(x + \frac{\left((\kappa_\parallel - \kappa_\perp)y sin(\theta)cos(\theta)\right)}{\left(\kappa_\perp cos^2(\theta) + \kappa_\parallel sin^2(\theta)\right)}\right)^2 \\
&+ \left(\left(\tfrac{\kappa_\perp}{2}sin^2(\theta) + \tfrac{\kappa_\parallel}{2}cos^2(\theta)\right) - \frac{(\kappa_\parallel - \kappa_\perp)^2 sin^2(\theta)cos^2(\theta)}{2\left(\kappa_\perp cos^2(\theta) + \kappa_\parallel sin^2(\theta)\right)}\right)y^2
\end{aligned}
\tag{46}
$$

Now looking just at the $y^2$ portion, we can determine an effective bending coefficient in the $\hat{y}$ direction.

$$
\begin{aligned}
\beta E_{\text{bend}}^{y^2} &= \tfrac{1}{2}\left(\left(\bar{\kappa}_\perp sin^2(\theta) + \bar{\kappa}_\parallel cos^2(\theta)\right) - \frac{(\bar{\kappa}_\parallel - \bar{\kappa}_\perp)^2 sin^2(\theta)cos^2(\theta)}{\left(\bar{\kappa}_\perp cos^2(\theta) + \bar{\kappa}_\parallel sin^2(\theta)\right)}\right)y^2 \\
&= \tfrac{1}{2}\left(\frac{\bar{\kappa}_\perp^2 cos^2(\theta)sin^2(\theta) + \bar{\kappa}_\parallel^2 cos^2(\theta)sin^2(\theta) + \bar{\kappa}_\parallel\bar{\kappa}_\perp\left(cos^4(\theta) + sin^4(\theta)\right)}{\bar{\kappa}_\perp cos^2(\theta) + \bar{\kappa}_\parallel sin^2(\theta)} - \frac{(\bar{\kappa}_\parallel - \bar{\kappa}_\perp)^2 sin^2(\theta)cos^2(\theta)}{\bar{\kappa}_\perp cos^2(\theta) + \bar{\kappa}_\parallel sin^2(\theta)}\right)y^2 \\
&= \tfrac{1}{2}\left(\frac{\bar{\kappa}_\parallel\bar{\kappa}_\perp\left(cos^4(\theta) + sin^4(\theta)\right) + 2\bar{\kappa}_\parallel\bar{\kappa}_\perp sin^2(\theta)cos^2(\theta)}{\bar{\kappa}_\perp cos^2(\theta) + \bar{\kappa}_\parallel sin^2(\theta)}\right) \\
&= \tfrac{1}{2}\left(\frac{\bar{\kappa}_\parallel\bar{\kappa}_\perp\left(cos^2(\theta) + sin^2(\theta)\right)^2}{\bar{\kappa}_\perp cos^2(\theta) + \bar{\kappa}_\parallel sin^2(\theta)}\right)y^2 \\
&= \tfrac{1}{2}\left(\frac{\bar{\kappa}_\parallel\bar{\kappa}_\perp}{\bar{\kappa}_\perp cos^2(\theta) + \bar{\kappa}_\parallel sin^2(\theta)}\right)y^2 \\
&= \tfrac{1}{2}\bar{\kappa}_{\text{Eff}} y^2
\end{aligned}
\tag{47}
$$

Because $\bar{\kappa}_\perp < \bar{\kappa}_\parallel$, $\kappa_{\text{Eff}}$ has a maximum at $\theta = 0$ and a minimum at $\theta = \tfrac{\pi}{2}$. In the limit where $\bar{\kappa}_\perp << \bar{\kappa}_\parallel$, we find

$$
\begin{aligned}
\beta E_{\text{bend}}^{y^2} &= \tfrac{1}{2}\left(\frac{\bar{\kappa}_\perp}{sin^2(\theta)}\right)y^2 \\
&= \tfrac{1}{2}\kappa_{\text{Eff}} y^2
\end{aligned}
\tag{48}
$$

where $\kappa_{\text{Eff}} = \frac{3l_p k_B T}{L^3 sin^2(\theta)}$, agreeing with previous models (**Mogilner and Oster, 1996**, BiophysJ), except for the fact that we arrive at a prefactor of 3 rather than 4, as we took into account all of the bending modes of the filament, rather than assuming bending of the filament lies along an arc of constant curvature - a difference also arrived at by Dickinson and colleagues (Dickinson, Caro, and Purich, 2004, BiophysJ).

This gives us a final bending energy…

$$
\beta E_{\text{bend}} = \tfrac{1}{2}\left(\frac{\bar{\kappa}_\perp \cdot \bar{\kappa}_\parallel}{\bar{\kappa}_{\text{Eff}}}\right)(x + f(y))^2 + \tfrac{1}{2}\bar{\kappa}_{\text{Eff}} y^2
\tag{49}
$$

## Calculation of the partition function

Using these energies, we can determine the partition function, $Z_{conf}$, summing over all possible positions of the filament. In this case, the filament can fluctuate freely along the x-axis, but cannot fluctuate past the membrane position along the y-axis. Here we define the membrane position $d$ relative to the filament tip.

$$
\begin{aligned}
Z_{conf} &= \int_{-\infty}^{\infty}\int_{-\infty}^{d} e^{-\beta E_{bend}}\, dy\, dx \\
&= \int_{-\infty}^{\infty}\int_{-\infty}^{d} e^{-\left(\frac{1}{2}\frac{\bar{\kappa}_{\perp}\cdot\bar{\kappa}_{\parallel}}{\bar{\kappa}_{\text{Eff}}}\right)(x+f(y))^2 - \left(\frac{1}{2}\bar{\kappa}_{\text{Eff}}y^2\right)}\, dy\, dx \\
&= \int_{-\infty}^{d} \sqrt{\frac{\pi}{\left(\frac{1}{2}\frac{\bar{\kappa}_{\perp}\cdot\bar{\kappa}_{\parallel}}{\bar{\kappa}_{\text{Eff}}}\right)}}\, e^{-\frac{1}{2}\bar{\kappa}_{\text{Eff}}y^2}\, dy \\
&= \sqrt{\frac{2\pi\bar{\kappa}_{\text{Eff}}}{\bar{\kappa}_{\perp}\cdot\bar{\kappa}_{\parallel}}}\int_{-\infty}^{d} e^{-\frac{1}{2}\bar{\kappa}_{\text{Eff}}y^2}\, dy
\end{aligned}
\tag{50}
$$

If we then have a change of variables, where we center the system at d, the equation becomes…

$$
Z_{conf} = \sqrt{\frac{2\pi\bar{\kappa}_{\text{Eff}}}{\bar{\kappa}_{\perp}\cdot\bar{\kappa}_{\parallel}}}\int_{0}^{\infty} e^{-\frac{1}{2}\bar{\kappa}_{\text{Eff}}(y-y_0)^2}\, dy
\tag{51}
$$

where $y_0$ is the equilibrium filament position, measured relative to the membrane surface (the filament pokes through the membrane for $y_0 < 0$).

## Calculation of the force of actin on a membrane

Upon an infinitesimal membrane position perturbation $\xi$, the equilibrium filament tip position becomes $y_0 \to y_0 + \xi$. The partition function becomes…

$$
\begin{aligned}
Z_{conf}(\xi) &= \sqrt{\frac{2\pi\bar{\kappa}_{\text{Eff}}}{\bar{\kappa}_{\perp}\cdot\bar{\kappa}_{\parallel}}}\int_{0}^{\infty} e^{-\frac{1}{2}\bar{\kappa}_{\text{Eff}}(y-y_0-\xi)^2}\, dy \\
&= \sqrt{\frac{2\pi\bar{\kappa}_{\text{Eff}}}{\bar{\kappa}_{\perp}\cdot\bar{\kappa}_{\parallel}}}\int_{-\xi}^{\infty} e^{-\frac{1}{2}\bar{\kappa}_{\text{Eff}}(y'-y_0)^2}\, dy'
\end{aligned}
\tag{52}
$$

and the derivative of the partition function with respect to the perturbation is (by Leibniz's rule)…

$$
\begin{aligned}
\left.\frac{\partial \tilde{Z}_{conf}}{\partial \xi}\right|_{\xi=0} &= \left.\frac{\partial}{\partial \xi}\left(\sqrt{\frac{2\pi\bar{\kappa}_{\text{Eff}}}{\bar{\kappa}_{\perp}\cdot\bar{\kappa}_{\parallel}}}\int_{-\xi}^{\infty} e^{-\frac{1}{2}\bar{\kappa}_{\text{Eff}}(y'-y_0)^2}\, dy'\right)\right|_{\xi=0} \\
&= \left.-\sqrt{\frac{2\pi\bar{\kappa}_{\text{Eff}}}{\bar{\kappa}_{\perp}\cdot\bar{\kappa}_{\parallel}}}\, e^{-\frac{1}{2}\bar{\kappa}_{\text{Eff}}(-\xi-y_0)^2}\right|_{\xi=0} \\
&= -\sqrt{\frac{2\pi\bar{\kappa}_{\text{Eff}}}{\bar{\kappa}_{\perp}\cdot\bar{\kappa}_{\parallel}}}\, e^{-\frac{1}{2}\bar{\kappa}_{\text{Eff}}y_0^2}
\end{aligned}
\tag{53}
$$

Plugging Appendix *Equation 51* and *Equation 53* into Appendix *Equation 5*, we get the following force:

$$
\begin{aligned}
F_{actin} &= \left.\frac{\partial G}{\partial \xi}\right|_{\xi=0} \\
&= -\frac{k_B T}{\tilde{Z}_{conf}}\left.\left(\frac{\partial \tilde{Z}_{conf}}{\partial \xi}\right)\right|_{\xi=0} \\
&= -\frac{k_B T}{\tilde{Z}_{conf}}\left(-\sqrt{\frac{2\pi\bar{\kappa}_{\text{Eff}}}{\bar{\kappa}_{\perp}\cdot\bar{\kappa}_{\parallel}}}\, e^{-\frac{1}{2}\bar{\kappa}_{\text{Eff}}y_0^2}\right) \\
&= \frac{k_B T e^{-\frac{1}{2}\bar{\kappa}_{\text{Eff}}y_0^2}}{\int_{0}^{\infty} e^{-\frac{1}{2}\bar{\kappa}_{\text{Eff}}(y-y_0)^2}\, dy}
\end{aligned}
\tag{54}
$$

Note that this equation for the force assumes that the pointed (non-growing) ends of the filaments (as well as branches) are rigidly fixed to a stiff and immobile actin network.

## Detailed Derivations IV: Numerical approximations

In this section, we describe the various numerical approximations made in the development of our computational model.

### Membrane stretch/bend forces

For membrane stretch/bend force calculations, the 2nd and 4th order spatial derivatives were calculated using a central finite difference approximation with 8th order accuracy.

## Force of actin on the membrane

Many equations used in this model required performing numerical calculations of Gaussians over half-space. For this purpose, we used the error function. The denominator of Appendix *Equation 54* can be rewritten in terms of the error function in the following way…

$$
\begin{aligned}
f_{\text{denominator}} \quad &= \int_0^\infty dy \, e^{-\frac{1}{2}\bar{\kappa}_{\text{Eff}}(y-y_0)^2} \\
&= \int_{y_0}^{-\infty} (-d\bar{y}) \, e^{-\frac{1}{2}\bar{\kappa}_{\text{Eff}}\bar{y}^2} \\
&= \int_{-\infty}^{y_0} d\bar{y} \, e^{-\frac{1}{2}\bar{\kappa}_{\text{Eff}}\bar{y}^2} \\
&= \int_{-\infty}^{0} d\bar{y} \, e^{-\frac{1}{2}\bar{\kappa}_{\text{Eff}}\bar{y}^2} + \int_0^{y_0} d\bar{y} \, e^{-\frac{1}{2}\bar{\kappa}_{\text{Eff}}\bar{y}^2} \\
&= \left[ \left( \frac{1}{2}\sqrt{\frac{\pi}{(\frac{1}{2}\bar{\kappa}_{\text{Eff}})}} \right) \right] + \frac{1}{2}\sqrt{\frac{\pi}{\frac{1}{2}\bar{\kappa}_{\text{Eff}}}} \text{Erf}\left[ \sqrt{\frac{\bar{\kappa}_{\text{Eff}}}{2}} y_0 \right] \\
&= \sqrt{\frac{\pi}{2\bar{\kappa}_{\text{Eff}}}} \left( 1 + \text{Erf}\left[ \sqrt{\frac{\bar{\kappa}_{\text{Eff}}}{2}} y_0 \right] \right)
\end{aligned}
\tag{55}
$$

The error function appears in the equation as $1 + \text{erf}(z)$, which can give inaccurate calculations when $z << 0$ and $1 + \text{erf}(z) \approx 0$. Therefore, in the low $z$ regime ($z < 0$), we replaced $1 + \text{erf}(z)$ with $\text{erfc}(|z|)$. (Because $\text{erf}(-z) = -\text{erf}(z)$, $\text{erf}(z(z < 0)) = -\text{erf}(|z(z < 0)|)$). It follows that $1 + \text{erf}(z(z < 0)) = 1 - \text{erf}(|z(z < 0)|) = \text{erfc}(|z(z < 0)|)$. The following cases are listed below for the relevant equations.

$$
F_{actin} = \begin{cases} y_0 > 0, & \dfrac{k_B T e^{-\frac{1}{2}\bar{\kappa}_{\text{Eff}}y_0^2}}{\sqrt{\frac{\pi}{2\bar{\kappa}_{\text{Eff}}}}\left(1+\text{erf}\left[\sqrt{\frac{\bar{\kappa}_{\text{Eff}}}{2}}y_0\right]\right)} \\[20pt] y_0 \leq 0, & \dfrac{k_B T e^{-\frac{1}{2}\bar{\kappa}_{\text{Eff}}y_0^2}}{\sqrt{\frac{\pi}{2\bar{\kappa}_{\text{Eff}}}}\text{erfc}\left(\left|\sqrt{\frac{\bar{\kappa}_{\text{Eff}}}{2}}y_0\right|\right)} \end{cases}
\tag{56}
$$

When the error function calculations failed (e.g., produced values of 0 or $\infty$), variable precision accuracy was used.

## Probability of adding a monomer

We used similar error function approximations to calculate the probabilities that thermal fluctuations of the filament open up a gap between the filament tip and the membrane of sufficient size to add a monomer: (See Detailed Derivations IV: Numerical approximations, Force of actin on the membrane for rational on using the erf and erfc functions.)

$$
p_{gap} = \begin{cases} y_0 > \Delta, & \dfrac{\left(1+\text{Erf}\left[\sqrt{\frac{\kappa_{\text{Eff}}}{2}}(y_0-\Delta)\right]\right)}{\left(1+\text{Erf}\left[\sqrt{\frac{\kappa_{\text{Eff}}}{2}}y_0\right]\right)} \\[20pt] \Delta \geq y_0 > 0, & \dfrac{\text{erfc}\left(\left|\sqrt{\frac{\bar{\kappa}_{\text{Eff}}}{2}}(y_0-\Delta)\right|\right)}{\left(1+\text{Erf}\left[\sqrt{\frac{\kappa_{\text{Eff}}}{2}}y_0\right]\right)} \\[20pt] y_0 \leq 0, & \dfrac{\text{erfc}\left(\left|\sqrt{\frac{\bar{\kappa}_{\text{Eff}}}{2}}(y_0-\Delta)\right|\right)}{\text{erfc}\left(\left|\sqrt{\frac{\bar{\kappa}_{\text{Eff}}}{2}}y_0\right|\right)} \end{cases}
\tag{57}
$$

When the error function calculations failed (e.g., produced values of 0 or $\infty$), variable precision accuracy was used.

## Choice of timestep

The timestep was chosen to capture the fastest dynamics in the system, which could either be the membrane stretch/bend relaxation, or actin network growth dynamics. This was implemented as $\min(\frac{0.1}{r_{max}^{actin}}, \frac{1}{r_{max}^{memb}})$. This timestep was calculated specifically for each simulation, depending on the parameters chosen. The timescales of the actin dynamics are set by the rates of polymerization, depolymerization, branching, and capping. To estimate the timescales of relaxation for membrane elastic forces, we calculate the relaxation timescales of the membrane fluctuating under Brownian thermal forces. (We do this because we do not a priori have an analytical theory describing the shape profile of the actin dynamics. Importantly, the fact that our simulations are not affected by the timestep (Fig. S3) provide evidence that we are sufficiently resolving all system dynamics using this

approximation to set the timestep.) For a membrane at low Reynolds number, we have the following equation of motion.

$$0 = -\gamma \frac{\partial y}{\partial t} + \sigma_{\text{eff}} \frac{\partial^2 y}{\partial x^2} - \kappa_{\text{eff}} \frac{\partial^4 y}{\partial x^4} + \xi(t)$$
$$\frac{\partial y}{\partial t} = \frac{1}{\gamma} \left( \sigma_{\text{eff}} \frac{\partial^2 y}{\partial x^2} - \kappa_{\text{eff}} \frac{\partial^4 y}{\partial x^4} + \xi(t) \right) \tag{58}$$

The random Brownian force is Gaussian distributed with mean $\mu = 0$ and variance $\sigma^2 = 2\gamma k_B T$ and is defined by its autocorrelation function

$$\langle \xi(x,t)\xi(x',t') \rangle = 2\gamma k_B T \delta(t - t')\delta(x - x') \tag{59}$$

We are ultimately interested in the time evolution of wave mode solutions to this equation, so we first take the Fourier transform in order to solve for the amplitude of each wavemode. Using the following properties of the Fourier transform:

$$\mathcal{F}(f(x,t)) = F(k,t) = \sqrt{\frac{1}{L}} \sum_{k=1}^{\infty} \cos\left( \frac{2\pi kx}{L} \right) f(x,t)$$

$$\mathcal{F}\left( \frac{d^n f(x,t)}{dx^n} \right) = \left( \frac{i2\pi k}{L} \right)^n \mathcal{F}(f(x,t))$$

$$\mathcal{F}\left( \frac{d^n f(x,t)}{dt^n} \right) = \frac{d^n}{dt^n} \mathcal{F}(f(x,t))$$

we can re-write the equation in Fourier space as…

$$\begin{aligned} \frac{dY(k,t)}{dt} &= \frac{1}{\gamma} \left( \sigma_{\text{eff}} \left( \frac{i2\pi k}{L} \right)^2 Y(k,t) - \kappa_{\text{eff}} \left( \frac{i2\pi k}{L} \right)^4 Y(k,t) + \Xi(k,t) \right) \\ &= -\frac{Y(k,t)}{\gamma} \left( \sigma_{\text{eff}} \left( \frac{2\pi k}{L} \right)^2 + \kappa_{\text{eff}} \left( \frac{2\pi k}{L} \right)^4 \right) + \frac{\Xi(k,t)}{\gamma} \\ &= -Y(k,t) \left( \frac{\sigma_{\text{eff}}\alpha}{\gamma} + \frac{\kappa_{\text{eff}}\alpha^2}{\gamma} \right) + \frac{\Xi(k,t)}{\gamma} \end{aligned} \tag{60}$$

where $\alpha = \frac{2\pi k}{L}$. The random Brownian force in Fourier space is Gaussian distributed with mean $\mu = 0$ and variance $2\gamma k_B T \Delta x$ as defined by its autocorrelation function. For a discrete system, a Fourier transform with normalization $\frac{1}{\sqrt{N}}$, where N is the number of discrete membrane segments in this case, preserves the variance and standard deviation of a vector of normally distrbuted random values. Here, our normalization is $\frac{1}{\sqrt{L}}$ in addition to integrating over the fixed segment length $\Delta x$, giving a final normalization of $\sqrt{\frac{\Delta x}{N}}$ for the Brownian force $\Xi(k,t)$. The factor of $\frac{1}{\sqrt{N}}$ preserves the variance, leaving the variance changed only by the multiplicative factor of $(\sqrt{\Delta x})^2 = \Delta x$.

$$\langle \Xi(k,t)\Xi(k',t') \rangle = 2\gamma k_B T \Delta x \delta(t - t')\delta(k - k') \tag{61}$$

We can solve this equation using a Laplace transform.

$$s\hat{Y}(k,s) - Y(k,0) = -\hat{Y}(k,s) \left( \frac{\sigma_{\text{eff}}\alpha}{\gamma} + \frac{\kappa_{\text{eff}}\alpha^2}{\gamma} \right) + \frac{\hat{\Xi}(k,s)}{\gamma} \tag{62}$$

$$\hat{Y}(k,s) = \frac{\frac{\hat{\Xi}(k,s)}{\gamma} + Y(k,0)}{s + \left( \frac{\sigma_{\text{eff}}\alpha}{\gamma} + \frac{\kappa_{\text{eff}}\alpha^2}{\gamma} \right)} \tag{63}$$

and then in inverse Laplace transform

$$\begin{aligned} Y(k,t) &= \int_0^t e^{-\left( \frac{\sigma_{\text{eff}}\alpha}{\gamma} + \frac{\kappa_{\text{eff}}\alpha^2}{\gamma} \right)(t-\tau)} \frac{\Xi(k,\tau)}{\gamma} d\tau + Z(k,0) e^{-\left( \frac{\sigma_{\text{eff}}\alpha}{\gamma} + \frac{\kappa_{\text{eff}}\alpha^2}{\gamma} \right)t} \\ &= \int_0^t e^{-\frac{\Gamma}{\gamma}(t-\tau)} \frac{\Xi(k,\tau)}{\gamma} d\tau + Y(k,0) e^{-\frac{\Gamma}{\gamma}t} \end{aligned} \tag{64}$$

Using the properties

$$\begin{aligned}
\mathcal{L}[(f*g)(t)] &= \mathcal{L}\left[\int_0^t f(\tau)g(t-\tau)d\tau\right] \\
&= F(s) \cdot G(s)
\end{aligned} \tag{65}$$

and

$$\mathcal{L}[af(t)] = aF(s) \tag{66}$$

and

$$\mathcal{L}[e^{-at} \cdot u(t)] = \frac{1}{s+a} \tag{67}$$

and

$$\mathcal{L}[f'(t)] = sF(s) - f(0) \tag{68}$$

Now that we have $Y(k,t)$, we can solve for for the time-autocorrelation function $(k',t)Y(k,0)\rangle$.

$$\begin{aligned}
\langle Y(k',t)Y(k,0)\rangle &= \left\langle\left[\int_0^t e^{-\frac{\Gamma}{\gamma}(t-\tau)}\frac{\Xi(k',\tau)}{\gamma}d\tau + Z(k',0)e^{-\frac{\Gamma}{\gamma}t}\right]Y(k,0)\right\rangle \\
&= \langle Y(k',0)Y(k,0)\rangle e^{-\frac{\Gamma}{\gamma}t}
\end{aligned} \tag{69}$$

We know that $(k',0)Y(k,0)\rangle$ is non-zero only if $k = k'$.

$$(k',0)Y(k,0)\rangle = \lim_{t\to\infty}(k,t)Y(k,t)\rangle \tag{70}$$

$$\begin{aligned}
\lim_{t\to\infty}\langle Y(k,t)Y(k,t)\rangle &= \lim_{t\to\infty}[\langle Y(k,t)\rangle]^2 \\
&= \lim_{t\to\infty}\left[\left\langle\int_0^t e^{-\frac{\Gamma}{\gamma}(t-\tau)}\frac{\Xi(k',\tau)}{\gamma}d\tau + Z(k,0)e^{-\frac{\Gamma}{\gamma}t}\right\rangle\right]^2 \\
&= \lim_{t\to\infty}\left[\left\langle\int_0^t e^{-\frac{\Gamma}{\gamma}(t-\tau)}\frac{\Xi(k',\tau)}{\gamma}d\tau\right\rangle\right]^2 \\
&= \lim_{t\to\infty}\int_0^t\int_0^t\left\langle e^{-\frac{\Gamma}{\gamma}(t-\tau)}e^{-\Gamma(t-\tau')}\frac{\Xi(k',\tau)}{\gamma}\frac{\Xi(k',\tau')}{\gamma}\right\rangle d\tau'd\tau \\
&= \frac{1}{\gamma^2}\lim_{t\to\infty}\int_0^t\int_0^t e^{-\frac{\Gamma}{\gamma}(t-\tau)}e^{-\frac{\Gamma}{\gamma}(t-\tau')}\langle\Xi(k,\tau)\Xi(k,\tau')\rangle d\tau'd\tau' \\
&= \frac{1}{\gamma^2}\lim_{t\to\infty}\int_0^t\int_0^t e^{-\frac{\Gamma}{\gamma}(t-\tau)}e^{-\frac{\Gamma}{\gamma}(t-\tau')}\left(2\gamma k_B T\Delta x\delta(\tau'-\tau)\right)d\tau'd\tau \\
&= \frac{2\gamma k_B T\Delta x}{\gamma^2}\lim_{t\to\infty}\int_0^t e^{-\frac{\Gamma}{\gamma}(t-\tau)}\left(\int_0^t e^{-\frac{\Gamma}{\gamma}(t-\tau')}\delta(\tau'-\tau)d\tau'\right)d\tau \\
&= \frac{2k_B T\Delta x}{\gamma}\lim_{t\to\infty}\int_0^t e^{-\frac{\Gamma}{\gamma}(t-\tau)}e^{-\frac{\Gamma}{\gamma}(t-\tau)}d\tau \\
&= \frac{2k_B T\Delta x}{\gamma}\lim_{t\to\infty}\int_0^t e^{-2\frac{\Gamma}{\gamma}(t-\tau)}d\tau \\
&= \frac{2k_B T\Delta x}{\gamma}\lim_{t\to\infty}e^{-2\frac{\Gamma}{\gamma}t}\int_0^t e^{2\frac{\Gamma}{\gamma}\tau}d\tau \\
&= \frac{k_B T\Delta x}{\frac{\Gamma}{\gamma}\gamma}\lim_{t\to\infty}e^{-2\frac{\Gamma}{\gamma}t}\left[e^{2\frac{\Gamma}{\gamma}t}-1\right] \\
&= \frac{k_B T\Delta x}{\frac{\Gamma}{\gamma}\gamma} \\
&= \frac{k_B T\Delta x}{\Gamma} \\
&= \frac{k_B T\Delta x}{\sigma_{\text{eff}}\lambda + \kappa_{\text{eff}}\lambda^2} \\
&= \frac{k_B T\Delta x}{\sigma_{\text{eff}}\left(\frac{2\pi k}{L}\right)^2 + \kappa_{\text{eff}}\left(\frac{2\pi k}{L}\right)^4} \\
&= \frac{k_B T}{\sigma h\left(\frac{2\pi k}{L}\right)^2 + \kappa h\left(\frac{2\pi k}{L}\right)^4}
\end{aligned} \tag{71}$$

by a variant of Fubini's theorem

$$\left(\int_a^b f(x)dx\right)^2 = \int_a^b f(x)f(y)dxdy$$

and a property of the Delta function

$$\int_{-\infty}^{\infty} f(x)\delta(x-a)dx = f(a)$$

So finally...

$$\langle k', t) Y(k, 0) \rangle = \frac{k_B T \delta_{kk'}}{\sigma h \left(\frac{2\pi k}{L}\right)^2 + \kappa h \left(\frac{2\pi k}{L}\right)^4} e^{-\left(\frac{\sigma_{\text{eff}}}{\gamma}\left(\frac{2\pi k}{L}\right)^2 + \frac{\kappa_{\text{eff}}}{\gamma}\left(\frac{2\pi k}{L}\right)^4\right)t}$$

(72)

From this equation, the rate of relaxation due to membrane elasticity is...

$$r^{\text{memb}} = \frac{\sigma}{\gamma}\left(\frac{2\pi k}{L}\right)^2 + \frac{\kappa}{\gamma}\left(\frac{2\pi k}{L}\right)^4$$

$$r^{\text{memb}}_{max} = \frac{\sigma}{\gamma}\left(\frac{2\pi N_{\text{segments}}}{L}\right)^2 + \frac{\kappa}{\gamma}\left(\frac{2\pi N_{\text{segments}}}{L}\right)^4$$

(73)

