## [Editor Report]

This paper describes analysis and modeling of leading edge fluctuations in migrating cells driven by a branched Arp2/3 lamellipodial network. A stochastic model shows how branching contributes to shape stability, and reproduces the measured spectrum and dynamics of leading edge fluctuations. Analysis of the model as a function of branching angle suggests that the Arp2/3 branching angle might be selected to smooth lamellipodial shape. This work provides new ideas to a big field of research, including Fourier analysis of leading edge fluctuations.

---

## [Decision Letter]

**Decision letter after peer review:**

Thank you for submitting your article "Leading edge maintenance in migrating cells is an emergent property of branched actin network growth" for consideration by *eLife*. Your article has been reviewed by two peer reviewers, including Alphee Michelot as the Reviewing Editor and Reviewer #1, and the evaluation has been overseen by Anna Akhmanova as the Senior Editor.

Essential revisions:

1) The main point raised by Reviewer #2 is closely related to the last point raised by previous Reviewer #3. The reviewer found that the analysis of Figure A1, A2 and A3 in the appeal to the other Journal shows that the red-blue stripe pattern of Figure 1b is indeed not intrinsic to the system. Presumably the features of Figure 1b reflect the length scale over which the curvature was calculated. Thus we agree that you should include the analysis of these figures in the manuscript and discuss these results to clarify these concerns.

2) Some explanations on spatial resolution would help (Reviewer #1). The spatial scales accessible with the experiments and those described in the model are very different. It would be useful to explain to which scales the experimental data remain informative for the understanding of these molecular mechanisms.

*Reviewer #1 (Recommendations for the authors):*

I have only one comment below which should be addressed before publication:

A missing point of this manuscript is the spatial resolution of these experiments based on phase-contrast microscopy. I find hard to believe that measured fluctuations arise from stochasticity in actin growth at the level of monomer addition, as claimed by the authors. This does not necessarily precludes all the conclusions from this study, but these seemingly exaggerated claims do not help readers understand the link between the experimental data and the model.

In a revised manuscript, I would suggest the authors to add a section addressing this problem of spatial resolution. This section should provide estimates of typical shape fluctuations that the experimental system is capable of detecting (or not), together with estimates of minimal network growth that can be detected. Next, the authors should indicate over which range of parameters their model can be compared with experimental data, and over which range of parameters their model becomes simply predictive.

*Reviewer #2 (Recommendations for the authors):*

To resolve my concern, the authors could plot the autocorrelation functions in Figure 1e at longer times and calculate Figure 1g using a smoothing length scale sufficiently larger than 7 microns.

---

## [Author Response]

Essential revisions:1) The main point raised by Reviewer #2 is closely related to the last point raised by previous Reviewer #3. The reviewer found that the analysis of Figure A1, A2 and A3 in the appeal to the other Journal shows that the red-blue stripe pattern of Figure 1b is indeed not intrinsic to the system. Presumably the features of Figure 1b reflect the length scale over which the curvature was calculated. Thus we agree that you should include the analysis of these figures in the manuscript and discuss these results to clarify these concerns.2) Some explanations on spatial resolution would help (Reviewer #1). The spatial scales accessible with the experiments and those described in the model are very different. It would be useful to explain to which scales the experimental data remain informative for the understanding of these molecular mechanisms.

We have now addressed both of these essential revisions in our updated manuscript. For our response to point (1), please see our additions to the Results (lines 121-128), Methods (lines 427-432, 452-454, 480-482, 497-512), and three new supplemental figures (Figure 1 —figure supplement 3-5). For our response to point (2), please see our expanded discourse on differences in spatial resolution between the experimental and theoretical aspects of our work in the Results (lines 92-99, 144-145, 153-154, 286-291) and Discussion (lines 318-325).

Reviewer #1 (Recommendations for the authors):I have only one comment below which should be addressed before publication:A missing point of this manuscript is the spatial resolution of these experiments based on phase-contrast microscopy. I find hard to believe that measured fluctuations arise from stochasticity in actin growth at the level of monomer addition, as claimed by the authors. This does not necessarily precludes all the conclusions from this study, but these seemingly exaggerated claims do not help readers understand the link between the experimental data and the model.In a revised manuscript, I would suggest the authors to add a section addressing this problem of spatial resolution. This section should provide estimates of typical shape fluctuations that the experimental system is capable of detecting (or not), together with estimates of minimal network growth that can be detected. Next, the authors should indicate over which range of parameters their model can be compared with experimental data, and over which range of parameters their model becomes simply predictive.

We have now done this. Please see our expanded discourse on this topic in the Results and Discussion.

Reviewer #2 (Recommendations for the authors):To resolve my concern, the authors could plot the autocorrelation functions in Figure 1e at longer times and calculate Figure 1g using a smoothing length scale sufficiently larger than 7 microns.

We have now done this. Please see additions to the Results, Materials and methods, and in three new supplemental figures (Figure 1 —figure supplement 3-5).

[Editors' note: we include below the reviews that the authors received from another journal, along with the authors’ responses.]

Reviewer #1:In their manuscript Garner and Theriot take a perturbation free observational approach in combination with theoretical modelling to address the question why lamellipodia have a smooth structure. They employ neutrophil like cells migrating under agarose pads and obtain movies with very high time resolution and find that the edge fluctuates locally. Autocorrelations show that the local fluctuations show a characteristic relaxation pattern that can be modelled using minimal components of actin polymerisation dynamics and membrane interaction. A striking result was that the relaxation could be reproduced even without including lateral mechanical interaction with the plasma membrane. The authors propose a model where the relaxation is an emergent property of the actin network geometry. Due to the 70{degree sign} branchpoint connections the filaments slide sideways and thereby the adjacent segments catch up, as the filaments sliding in from the lagging segments cannot catch up with the leading membrane and fail to spawn nucleation sites (the only occur in association with the membrane). Varying branching angles they find that the 70{degree sign} Arp2/3 branch is optimal for providing this behavior.This is very elegant and deep work and an extremely useful framework to think about the crucial question why the branch junction is conserved around 70{degree sign}. Some questions should be addressed before publication.1. Neutrophils under agarose are certainly develop lamellipodia, however their hydrostatic components seem quite strong and even in the phase contrast movie of the untreated cell a bleb seems to occur. It would be good to control that fluctuations are really driven by polymerisation and not by hydrostatics. Blebbistatin treated cells should still migrate and it would be good to see that the fluctuations are not affected.

The reviewer raises an excellent point that hydrostatics may play a role in mediating leading edge shape fluctuations, and the strength of this effect remains an exciting area of study. Unfortunately, blebbistatin would almost certainly perturb actin turnover and thus free monomer concentration (which would lead to similar phenotype to that observed under Latrunculin B treatment), in addition to hydrostatic pressure. This experiment would therefore be unlikely to shed light on the relative roles of actin polymerization and hydrostatic pressure on leading edge fluctuation dynamics. Future modeling efforts which explicitly include hydrostatic pressure might allow predictions of fluctuation behavior in hydrostatics-driven or polymerization-driven regimes, but we consider this to be outside the scope of the current work. We have added clarification in the discussion that other cellular properties, including hydrostatics, may be acting in cells to additionally modulate leading edge shape dynamics, and that subsequent additions to the model could be used to further dissect the relative strength of these effects in diverse contexts (lines 299-302, 310).

2. As neutrophils have little to no retrograde flow of actin the following question arises: if the agarose pad has some structural irregularity the load on the lamellipodial tip would fluctuate as well and forcevelocity relation could also drive fluctuations. It would be satisfying to see that the model also holds in a situation without agarose overlay. The Theriot lab should have plenty of movies of e.g. fish keratocytes to check if the fluctuations are comparable.

We thank the reviewer for their intriguing question regarding the effects of agarose and comparative analyses of leading edge fluctuations in HL-60s cells and keratocytes. We were sufficiently intrigued by the reviewer’s suggestion that we collected a new, high-speed imaging dataset on keratocytes for the first time. We are happy to report that keratocytes, in the absence of an agarose pad overlay, indeed exhibit leading edge fluctuations qualitatively and quantitatively similar to those of HL-60 cells under confinement. We have added a new supplementary companion figure to Figure 1 (Figure S3) describing leading edge fluctuations in keratocytes, and discuss these results in the main text (lines 123-125). We believe this fully addresses the reviewer’s question on the effect of agarose on the fluctuations in HL-60s. We note the model itself is not cell type-specific, and so should generally apply regardless of experimental condition. Therefore it was indeed satisfying to see that the results of the model are congruent with experimental data from both HL-60 cells and keratocytes.

3. The authors propose that filaments laterally sliding in from the lagging segments cannot keep up with the membrane, fall behind and do not contribute to new branches. Therefore the lagging segments that receive additional filaments sliding in from the leading segment catch up while the initially leading segment thins out and slows. This is a plausible result of the model – but I have difficulties understanding that the filaments laterally arriving from the lagging parts cannot keep up. Following a force velocity curve they should speed up and keep contact with the membrane instead of falling behind and being capped.

The reviewer states that filaments following a force-velocity curve should speed up to keep contact with the membrane. However, this is not the case for lamellipodial actin networks, according to established theoretical and experimental work. Rather, force-velocity curves of branched actin networks as measured by AFM in keratocyte lamellipodia (Prass, Jacobson, Mogilner, and Radmacher, 2006) and in vitro (Parekh, Chaudhuri, Theriot, and Fletcher, 2005) exhibit a largely load-independent velocity in low load regimes (<1 nN/µm^2^). This is largely due to the fact that dendritically branched actin networks increase in actin density (via branching) in response to increased force to maintain a constant amount of force per filament. Lamellipodial actin networks, in which filaments are much shorter than their persistence length and much stiffer than the plasma membrane, operate in this low load regime (Mogilner and Oster, 1996), consistent with electron microscopy images of the keratocyte lamellipodium, in which filaments are observed to remain straight, rather than bend, under the load of the leading edge membrane (Svitkina and Borisy, 1999). Therefore we do not expect significant changes in the polymerization velocity due to small variations in the load (and in particular, we do not expect lagging filaments to speed up). Rather, we expect actin density to decrease in response to a reduced load to maintain a constant polymerization velocity, which is consistent with our model. Thus there is no discrepancy between our model and the known force-velocity relationship.

4. Can the sideway sliding be directly seen as lateral propagation in in the curvature plots (Figure 1b)? When the red area splits the resulting Y junction should have a characteristic angle. Is this the case and if yes is it in quantitative agreement with the model?

We thank the reviewer for their intriguing observation of lateral propagation in the curvature kymograph. The filament spreading invoked by the model and schematized in Figure 4j should, as far as we know, act uniformly in both directions rather than slide unidirectionally. Although sliding should not be unidirectional, actin density in cells tends to be highest in the center of the leading edge and decreases towards the edges. Thus we might imagine spreading could manifest as curvature events that slide from the center towards the sides of the cell. However, spreading in our model occurs at the single-filament level. It remains an open area of research how this molecular-scale spreading phenomenon should propagate to the larger length-scales seen in the curvature kymograph – and further what implications that might have for lateral spreading of micron-scale curvature events. We remain excited about this future direction but find it to be outside of the scope of the current work.

Reviewer #2:In this work, the authors study the protrusion of membranes by polymerizing a branched actin network. In particular, the address the question of how a flat membrane configuration can be stable in presence of fluctuations due to molecular noise in the actin polymerization and branching process. They find that high actin density regions, which would locally deform the leading edge, are dissolved because newly nucleated filaments (daughter filaments) branch off under an angle from existing filaments (mothers). Their simulations indicate that the experimentally observed branching angle of 70 degrees is optimal for stabilizing a flat leading edge.The problem as such is interesting – analyzing the fluctuations of a membrane that is pushed by a polymerizing actin network – and also the mechanism by which fluctuations relax is nicely worked out but the authors, but I have several issues with this work that, in my opinion, preclude its publication in the Journal. First of all, the main motivation for the study is "Because the average cell shape remains constant over time, there must be some form of feedback acting on leading edge curvature to sustain stable lamellipodial growth." (ll 83) This is not necessarily true – and sure enough the authors find that no feedback is necessary, which they sell as a "surprising revelation" (l 278). There is a large body of literature in physics that studies surface growth and there it is very common to see that the roughness of the surface saturates with time, see for example, the ballistic deposition model. Also in the context of a polymerizing branched network this is at least implicitly known, see Weichsel and Schwarz, PNAS 107, 6305 (2010).

While the reviewer correctly states that many general physical models of surface growth exhibit a surface roughness that saturates with time, this is not necessarily the case for all systems, and certainly has not been established for lamellipodial actin. In particular, several features of lamellipodial actin are known to lead to nonlinearities in network growth, which might amplify stochastic fluctuations. First, dendritic branching is an autocatalytic process which can lead to explosive growth. Second, the “deposition” of actin is dependent on the velocity of the flexible membrane surface it is pushing, in a manner which imparts hysteresis to the system (Parekh et al., 2005). Finally, the geometry of actin network growth exhibits nontrivial interactions with the motion and shape of the leading edge membrane. It is far from obvious whether these generic surface growth models would capture each of these complexities. The reviewer cites Weichsel and Schwarz, PNAS 107, 6305 (2010), but this work only investigates actin network growth against a flat rigid surface – in which “surface roughness” is not defined and therefore does not provide a relevant point of comparison. Further, our spreading-null simulations (Figure 4f-i) explicitly demonstrate a case in which actin network growth exhibits instability – providing a direct counterexample to the reviewer’s expectation from generic surface growth models. For all of these reasons, the existence of a class of generic surface growth models that exhibit stability in no way precludes the significance and novelty of our results.

Furthermore, decades of work in cell biology and cell biophysics have investigated the need for extrinsic feedback mechanisms to maintain leading edge stability. This has been considered a major open question in the field – until now. For instance, several influential reviews and perspectives dedicated to the subject have identified the potential roles of membrane tension (Diz-Muñoz, Fletcher, and Weiner, 2013; Sens and Plastino, 2015), curvature-sensing proteins (Zegers and Friedl, 2015), and a competition for membrane-associated free monomers (Mullins, Bieling, and Fletcher, 2018) in maintaining a single, stable lamellipodium – in addition to several influential papers outlining mechanisms based on force-dependent directional filament branching (Risca et al., 2012) and WAVE complex self-organization (Pipathsouk et al., 2019). The current work shows that none of these mechanisms are required for the generation and maintenance of a stable lamellipodium, and thus represents a major advance in the field as well as a surprising and novel finding.

We have made substantial changes to the Introduction which more clearly set up why our major finding is surprising and a significant contribution to the field, including rearranging the logical flow of the Introduction, as well as: 1) strengthening our language regarding the decades of prior literature assuming some sort of regulatory or mechanical must be required for leading edge stability (lines 43-47); and 2) highlighting sources of known nonlinearities in actin network growth that likely amplify stochastic fluctuations (lines 53-59).

My second issue is that the argument the authors give for the 70 degrees branching angle are not convincing to me. First of all, the minimum in relaxation times is not very pronounced. Second, why should evolution single out the network structure that minimizes surface roughness? Would there be any way to test the evolutionary advantage experimentally?

We assume that the reviewer’s reference to the “minimum in relaxation times” is meant to refer to the minimum in leading edge shape fluctuation *amplitude*, seen in Figure 5h, as the manuscript never references a minimum relaxation time. It is difficult to state in absolute terms whether this minimum is or is not “pronounced”. While it is unknown quantitatively how the effect size on leading edge fluctuation amplitude translates to a functional advantage in cell motility, and further unknown how a functional advantage on the cellular level conveys a fitness advantage to the organism, there is no reason to assume the downstream effects of this minimum are small. In general, a small perturbation to one component of a physical system can have large effects the behavior of the system as a whole – and this certainly holds for many cellular processes (e.g. signaling cascades). Specifically in the case of evolutionary selection, it has certainly been established that even phenotypes with very small relative fitness advantages can come to dominate a population, and thus act as a driver of evolution (Crow and Kimura, 1970; Imhof and Schlotterer, 2001) Therefore it is not possible to conclude that an apparently shallow minimum is not functionally or biologically significant. Further, this minimum observed in our work is the only evidence of any kind proposed in the literature addressing the striking evolutionary conservation of the Arp2/3-mediated branching angle, and thus represents a significant finding for the field.

The reviewer’s next question regarding *why* evolution might minimize surface roughness is an excellent one, and one that could only even be asked now that our work has identified this important potential mechanism. As discussed in the previous paragraph, the relationship between molecular mechanism, cellular function, and organismal fitness is extremely nontrivial and currently poorly understood – and thus beyond the scope of our work, but we certainly hope that our thought provoking results will inspire future work to address this important question.

My third issue is with the way the model is set up, notably the membrane part. The membrane bending energy is incorporated in a strange way. By themselves, the springs connecting the membrane segments do not lead to a bending energy. Only by constraining the motion of the segments along the y-direction, it comes into play in some way. But then this is also membrane tension, so it's quite confusing to me.Furthermore, if I understood correctly, thermal fluctuations of the membrane are not taken into account.These also depend on the membrane tension, which would thus affect the roughness of the surface. Continuing with the segments, their size corresponds to which physical parameter? Probably something like the membrane's persistence length. When changing the membrane tension, then the persistence length changes and thus the size of the segments. All these points remained unclear to me and I think that the description of the membrane is not physically sound.

The reviewer is incorrect to state that bending energy comes into play within our model implementation “only by constraining the motion of the segments along the y-direction”. Rather, the bending energy is incorporated as the 2^nd^ derivative of the membrane position (curvature) in the energy functional (Supplementary text equation 2). For clarity we now provide a succinct physical interpretation of our energy functional, including where bending appears, to the Supplementary Text corresponding to Equation 2. Perhaps the reviewer interpreted “connected by springs” more literally than intended, as the springs were meant to represent elastic coupling (i.e., resisting both extension and curvature) between the segments. For clarity, we have revised the text to replace “connected by springs” with “coupled elastically to each other” (line 137).

The reviewer is also incorrect to assume that membrane fluctuations would affect “surface roughness”. It has long been established that actin filament fluctuations, rather than membrane fluctuations, drive membrane motion and monomer incorporation into the network (Mogilner and Oster, 1996). Therefore we do not expect thermal fluctuations of the membrane to significantly affect leading edge shape fluctuations – thus it is physically sound to exclude them. We have added clarification to this extent in the supplementary text associated with Equation 1.

Finally, the reviewer is correct that the membrane segment size must be sufficiently short to resolve the smallest length scales at which there is significant bending. Indeed, we took this into account in developing our model. For a freely floating membrane fluctuating under Brownian motion, the appropriate segment size would be at least as small as the membrane’s persistence length to correctly resolve the membrane dynamics. We chose a membrane segment size well below the persistence length of this membrane “strip”. However, what we have modeled is not a membrane fluctuating under Brownian motion, but a membrane being driven by a very stiff actin network. It is possible, even likely, that the system as a whole would have a different effective persistence length than a membrane alone (e.g., the actin growth dynamics might suppress bending of the leading edge below a certain curvature). Lacking analytical theory describing this effective persistence length of the actinmembrane system, the only way to know that the relevant length scales have been sufficiently resolved is to vary the segment size and confirm that there is no effect on the dynamics. We directly demonstrated that the membrane segment size is sufficiently small by varying the membrane segment size and show quantitatively that the leading edge fluctuations are not affected (Figure S5). This was referenced in the main text (lines 165-166), and now is described more explicitly in the Methods section on Modeling (lines 478-479). Therefore we are convinced that our treatment of the membrane is physically sound.

In conclusion, in my opinion, the authors need to tone down their claims and provide more physical justification of their model. With such revisions, they could submit the work to a more specialized journal.Minor commentsll. 79 "reminiscent of an excitable system in which local instabilities in the leading edge emerge, grow, and then relax" This is not sufficient for describing it as an excitable system. Fluctuations always grow initially and – if the system is stable – relax, but not all systems subject to fluctuations are excitable.

The reference to excitable systems has been removed (line 88).

ll. 80 "Consistent with the idea that these fluctuations arise from stochasticity in actin growth, kymograph analysis of curvature and velocity (Figure 1b-c) showed that relatively long-lived curvature fluctuations are formed by the continual time-integration of very short-lived (sub-second, sub-micron) velocity fluctuations." I fear that I do not understand this sentence.

It is unclear which part of the sentence confused the reviewer, but we thank them for drawing our attention to some potential points of confusion which we have clarified in the text as outlined below.

As our manuscript was written for a broad audience, one function of this sentence is to act as a succinct reminder (for the benefit of readers lacking a strong physics or mathematics background) of the mathematical relationship between velocity, position, and curvature. As velocity is defined as the first time-derivative of position, the membrane position results from the continual time-integration of the velocity fluctuations (i.e., the velocity fluctuations add up over time to determine the membrane positional fluctuations). As curvature is defined as the second spatial-derivative of the membrane position, curvature fluctuations also arise via a time-integration of the velocity fluctuations. We have replaced the phrase “curvature fluctuations” with “shape fluctuations” to avoid inadvertently implying that curvature is directly determined as the time-integral of the velocity (line 93).

Additionally, this sentence serves to emphasize that the small sub-second, sub-micron scales of the velocity fluctuations – visually appearing uncorrelated to and acting at different scales than the membrane curvature – are consistent with fluctuations arising from stochastic actin polymerization at the level of monomer addition. Several clarifying points along these lines have been added to this sentence (lines 91-94).

Figure 1d: I am not sure what to learn from this sketch.

Figure 1d is intended to provide readers without a strong background in physics or mathematics with an intuition for the relaxation of shape fluctuations, a pictographic description of the wavelength, λ, and relaxation timescale, τ, as used throughout the paper. Based on our experience presenting this work to audiences with a biological background, we have found that it is helpful to include a cartoon representation of this commonly observed relationship between wavelength and relaxation timescale for elastic systems (see figure caption for Figure 1d).

Figure 1e,f: What are the wavelengths associated with the chosen Fourier modes?

The spatial frequencies are equally spaced between 0.22-0.62 µm^-1^ in 0.06 µm^-1^ intervals, corresponding to wavelengths between 1.6-4.5 µm. This information has been added to the legend for Figure 1.

Ll 109 "This general trend is shared by many physical systems with linear elastic constraints, such as idealized membranes and polymers freely fluctuating under Brownian motion, but can be contrasted with systems that have a dominant wavelength, as in the case of buckling or wrinkling of materials under compression." I do not understand the relation between buckling and fluctuations. The buckling instability can be discussed within the frame of linear elasticity (Landau-Lifshitz, vol 7, {section sign}21).

Buckling is invoked here merely as an example to give readers without a strong physics background some intuition for the kinds of systems that exhibit a dominant wavelength. We are not claiming any specific connection between buckling and fluctuations.

L 138 "To our great surprise" Depends on your experience… I would suggest to take it out.

As we have described in our response to the reviewer’s first major concern, the result that a simple actin network growth model can reproduce stable leading edge fluctuations, in the absence of any extrinsic sources of feedback, is indeed a surprising result in the field. Therefore we believe that, while our choice is a stylistic preference, the use of this phrase correctly indicates to readers less familiar with actin biology that this result was highly unexpected given the state of the field.

Ll 151 "It should be noted that the generation of the simulated data in Figure 2k-l did not involve any curve-fitting (and therefore no free parameters that could be fit) to the experimentally-measured autocorrelation dynamics." I do not understand. The relaxation times and fluctuation amplitudes should depend on parameters. I think what you want to state is that the parameters you have taken from the literature give results that agree decently with the experimental data.

We did not mean to imply that the simulation results do not depend on the input parameters. Rather we state that we did not use curve-fitting on the experimentally measured fluctuations to parameterize the simulations. In modeling of biological systems, it is common to parameterize models by fitting to the experimental data, so we needed to make clear this is not the case. We have clarified these points in the text (line 168).

Figure 3 "Minimal model correctly predicts response of HL-60 cells to drug treatment." The model does not predict anything – and "correct" is not defined. You show that the trends and the range of fluctuation amplitude and decay rate are similar.

We thank the reviewer for prompting us to clarify how we defined predictive power in this context. The title for Figure 3 uses the shorthand “correctly predicts”, which is intended to convey that the outputs of the simulation are congruent with experimental observations under conditions that had not been tested prior to model development. (Specifically, our model originally developed using parameters measured for unperturbed cells is able to recapitulate qualitative and quantitative features of the experimental data for latrunculin-treated cells, demonstrating how a reduction in free monomer concentration should affect quantitative parameters such as leading edge fluctuation amplitudes and timescales.) This is obviously too long for a figure title; however, to address the reviewer’s concern we now explicitly define how we assess the model’s predictive power in the main text associated with Figure 3 (lines 175-177).

Ll 233 "only a handful of mother filament angles (θf) survive until the end of the simulation" I do not understand – there is a well-defined average of θ_f_ and some variation around this angle; what does "handful of mother filament angles" mean? In Figure 5a there is no distinction between mother and daughter filaments, right?

We thank the reviewer for drawing our attention to this potential point of confusion. The reviewer is correct that there is no distinction between mother and daughter filaments in Figure 5a. In this sentence we used “mother filament[s]” to identify the filaments which seeded the simulation. Because there are a finite set of discrete filament angles represented in this randomized starting population, the “surviving” filament angles are descended from a few “ideal” seed filament angles. We agree this is slightly confusing, as any filament which creates a branch at any time in the simulation is technically a mother filament to its branch. For clarity, we have replaced “mother filament angles” with “the initial filament angles” (line 251).

Ll 234 "successful filament angles are narrowly (…) distributed around one half of the branching angle" What does "narrowly" mean? 'Narrow' is a relative notion, right?

We agree with the reviewer that “narrow” is a relative notion. However, the referenced distribution is narrow by almost all reasonable standards. The distribution is narrow relative to the initial population of seed filament angles, relative to the range of total possible angles (0°-180°), relative to the branching angle, and relative to the resolution of experimental measurements of lamellipodial filament angles by scanning electron microscopy (SEM).

Figure 5h,l, q: add labels to the x-axes.

It is unclear what the reviewer is referring to. The x-axes on Figure 5 h, l, and q are labeled. Possibly the reviewer is referring to the *insets* of Figure 5h,l,p? The figure legend states “(h,l,p) […] Insets have identical x-axes to main panels.” Adding duplicate, identical x-axis labels to the insets would only make the figure more cluttered and difficult to interpret.

Ll 293 "The defining characteristic and major advance of our model was the explicit inclusion of both the evolutionary dynamics of the actin network and its interaction with the two-dimensional geometry of the leading edge." But this the membrane is not essential, see also Weichsel and Schwarz op cit, who produced a similar network dynamics but without an explicit membrane.

As the reviewer correctly states, Weichsel and Schwarz 2010 did not include an explicit membrane – and therefore *could not possibly* have explored the interaction between network dynamics and membrane bending leading to the two-dimensional leading edge geometry (as we have done here). We have been careful to cite the landmark papers describing the evolutionary aspects of the actin network (Maly and Borisy, 2001; Schaus and Borisy, 2008; Schaus, Taylor, and Borisy, 2007), and we do not claim that our reproduction of these results is novel. Rather, it is the interaction between these evolutionary dynamics and the two-dimensional leading edge shape which is novel in our work.

Ll 368 "for which we used a conservative cut-off of 10 sec." I am not sure what this means. Could you present a graph showing the steady state has been reached after 10s?

Our model’s approach to steady state can be seen in three out of our five main text figures (Fid 2d-h, Figure 4a-e, and Figure 5a), as well as three out of our five supplementary figures (Figure S4c-g, Figure S5c-g, Figure S6c-g), and is directly referenced in the main text: “Nascent leading edges reach steady state values for filament density, filament length, filament angle, membrane velocity, and (most importantly) membrane fluctuation amplitude within seconds, a biologically realistic time scale (Figure 2d-h).” (lines 154-156). However, for good measure we have added a reference to Figure 2d-h at the end of the line referenced by the reviewer (line 423).

Ll 400 "We expanded this formalism to include all fluctuations of the filament along the filament's short and long axes." In ll 397 you write that filaments were fixed in position, now you write about fluctuations along the filament direction and perpendicular to it. Could you clarify?

As described in the work by Mogilner and Oster (1996) cited in this section, “filaments apply force to the membrane segments according to the untethered Brownian ratchet formalism^5^”. In this formalism, filaments are assumed to be rigidly attached to the actin network at their branch points. The filament thus fluctuates around this rigid attachment. This rigid attachment is fixed in position (in the lab frame of reference), such that filaments do not undergo retrograde flow (i.e. translation of the filament position opposite the direction of migration) or translational diffusion. Clarification of how filaments are translationally fixed in position at one end (lines 461-462), and freely fluctuating at the other end (lines 465-466) is now included.

Supplementary MaterialI do not understand Equation (3). F_S/B_ should be δES/Bδy– I am not sure, why you integrate over x. Furthermore, why is the integral over x replaced by multiplication with Δ x? Doesn't the integral extend over the whole membrane? – Now I have seen Equation (14), which clarifies things. I would suggest that you add the integration boundaries to Equation (3) to make it clearer.

As the reviewer correctly understood after having read the appendix referenced in the text corresponding to Equation 3, δES/Bδy is a functional derivative (as opposed to an ordinary derivative) having units of force per unit length, and must be integrated over the size of a discrete membrane segment in order to determine the force on said segment. We have added the integration boundaries as suggested, and added clarifications in the text associated with Eq 3 and Eq 14.

App. II: I still do not understand how connecting segments by springs lead to a bending energy. I guess that it is somehow related to the fact that the segments can only move along the y-direction.

Please see our response to the reviewer’s third major concern (above), where we address this question.

Reviewer #3:This paper describes analysis and modeling of leading edge fluctuations in migrating cells driven by a branched Arp2/3 lamellipodial network. A stochastic model shows how branching contributes to shape stability, thus explaining the measured spectrum and dynamics of leading edge fluctuations. Analysis of the model as a function of branching angle suggests that the Arp2/3 branching angle might be selected to smooth lamellipodial shape. The topic of actin driven cell motility will be of general interest and appropriate for the Journal. The authors provide new ideas to a big field of research, including Fourier analysis of leading edge fluctuations, which to my knowledge is a novel approach. The modeling methods and model design seem valid and appropriate to me and the paper is well written.I have however several questions on issues that are unclear to me and are central to the main results of the paper. The following comments focus on the validity of the analysis of experimental images to derive the observed spectrum of fluctuations, the interpretation of the model results, as well as whether the derived optimal Arp2/3 complex value may be dependent on the chosen parameter values.Major comments1) I have some questions and concerns regarding the analysis of leading edge behavior in Figure 1.Firstly, I am worried that looking at deviations of the leading edge contour away from a smoothed profile over 7 microns (as was done in this work) may be erasing the dynamics that cause the most significant fluctuations in the system. The behavior of the large wavelength fluctuations in Figure 1e-h may depend on this smoothing. Figure 1B shows a nearly periodic red-blue curvature pattern with a scale of order 4 microns that persists over 10-15 sec, a time over which the cell advances by a distance of order the size of the lamellipodium. Such a nearly periodic pattern would lead to a peak at the corresponding wavelength, which is not seen however in Figure 1g.

We appreciate the reviewer’s remarks on the details of our spectral analysis, which prompted us to incorporate several additions and clarifications to the manuscript text regarding the methods – including: 1) a detailed description on how the kymographs were generated, 2) a discussion on why kymographs fall short in characterizing fine-scale leading edge fluctuations, and 3) a discussion on how or whether the background subtraction is expected to affect the measured fluctuation amplitudes and dynamics. We believe these changes, in addition to our point-by-point response below, fully address the reviewer’s concerns.

With regards to the reviewer’s concern that the background subtraction removes “the most significant fluctuations in the system”, we emphasize that the most obvious fluctuations seen in the curvature kymograph are not necessarily the most significant. Indeed, the most obvious features in a kymograph will inevitably be the largest features occurring over the longest timescales. In contrast, the explicit goal of this work was to understand the fine-scale fluctuations in the leading edge shape that we discovered, and then took great pains to measure, using high-speed, high resolution imaging (Figure 1a). In fact, our work nicely demonstrates the limitations of kymograph analysis, where the larger-scale fluctuations are arbitrarily emphasized over fine-scale fluctuations – making obvious the need for a more quantitative treatment that encompasses all spatial scales, as exemplified by our Fourier autocorrelation analysis. We include the kymographs in Figure 1 to serve as a visual aid for readers who are most familiar with this kind of analysis to obtain a qualitative sense of the data, before quickly moving on to our more in-depth definitive analysis. To clarify these points for the reader, we have added to the Methods: 1) a new section describing the generation of the kymographs (lines 389-392) and 2) a discussion of our motivation for using spectral decomposition, including the limitations of kymograph analysis (lines 394-401).

In response to the reviewer’s concern that the background subtraction is removing oscillatory dynamics in the data, we do not believe a lack of oscillations in our Fourier analysis means that there is something wrong with our background subtraction technique. Most importantly, we do not see a mechanism by which *spatial* smoothing/filtering should remove *temporal* oscillations in our dataset. Even if for some reason the background subtraction were to partially suppress the 4 µm signal, it would do so uniformly in time, and therefore would maintain oscillations in the Fourier autocorrelation results. To elucidate these points, we have expanded our description of the spectrum analysis in the Methods to include an explicit discussion on the expected effects (or lack thereof) of the background subtraction on the measured fluctuation amplitudes and rates (lines 442-450).

Finally, we do not see evidence of strong temporal oscillations, as opposed to long-timescale fluctuations which appear and then decay, in our curvature kymograph. In general, it is non-trivial to prove the existence of oscillations in any noisy signal (e.g., see Liu, Oh, Peshkin, and Kirschner, 2020); certainly visual inspection of a kymograph does not suffice. We acknowledge some readers familiar with the field may be primed to look for oscillations due to their familiarity with protrusion-retraction patterns – which are not directly relevant to this system (please see our answer to the reviewer’s next comment for further discussion on this topic).

Therefore, our curvature kymograph is very much consistent with our Fourier autocorrelation results, and we have no reason to doubt our use of background subtraction for fluctuations larger than 7 µm.

Leading edge fluctuations over these spatial and temporal scales (microns, lamellipodium turnover time) have been considered in several prior works, including modeling studies by Zimmerman and Falcke, Ryan and Vavylonis, Gov group and others. The fact that there is no connection made to this body of work makes me wonder if the authors may have missed relevant questions answered or raised by these prior works.

We thank the reviewer for drawing attention to a potential point of confusion regarding the previously well-studied case of lamellipodial protrusion-retraction dynamics and their comparison to the leading edge shape fluctuations presented in this paper. The modeling studies mentioned by the reviewer only considered lamellipodia that undergo protrusion-retraction cycles, which is a fundamentally different mode of motility compared to that studied in this work – operating with distinct actin dynamics and biomechanics, exhibiting qualitatively different shape dynamics, and evolving over markedly slower (hundred-fold longer) timescales.

In particular, these cells cannot be undergoing protrusion-retraction cycles, because the leading edge does not exhibit any evidence of retraction (i.e. the velocity is never negative in Figure 1c). A line has been added to the caption for Figure 1c pointing out the uniformly positive velocity shown in the kymograph. In addition, cells that undergo protrusion-retraction-based motility, such as fibroblasts, typically migrate inefficiently, moving with average speeds ten- to a hundred-fold slower than HL-60 cells, and the oscillatory period of their protrusion-retraction dynamics is typically on the order of tens of *minutes*, while the fluctuations studied in this work maximally last tens of *seconds*. Therefore the reviewer is incorrect to say that the referenced prior publications considered leading edge fluctuations over the same temporal scales as our current work.

For all of these reasons, the leading edge fluctuations identified and explored in this work represent a novel class of leading edge dynamics, which are mechanistically distinct from protrusion-retraction cycles. To highlight the novelty of the observed dynamics, we have added a sentence to the manuscript clarifying how our measured leading edge fluctuations are distinct from the previously well-studied case of protrusion-retraction cycles, including citations to some of this literature (lines 89-91).

I think that the autocorrelation functions in Figure 1e should be shown at longer times to check that they decay to zero (or perhaps those with long λ may show weak periodicity as Figure 1b would suggest). It's not clear to me why none of the examples in Figure 1e do not decay significantly over time.

The reviewer’s assumption that the autocorrelation functions as plotted in Figure 1e should decay to zero is not correct, for reasons we now make more clear in the Methods (lines 432-436) and discuss here. The amplitude being plotted in Figure 1e is the complex magnitude of the autocorrelation values (this is stated in the figure legend as well as in the Methods). The Fourier transform and resulting autocorrelation values are complex, having both real and imaginary parts of the form A = a + ib. The complex magnitude of this value, sqrt(a^2^ + b^2^), is most representative of the total autocorrelation. We of course expect the autocorrelation to decay down to some noise window, which may be wavelength-dependent. Because we are plotting the complex magnitude (which is always positive), the noise window appears as a leveling-off of the autocorrelation curve to some non-zero background value. Therefore we do not expect the complex magnitude to decay down to zero, but rather to some finite background value, which is indeed what we find. The timescales shown in Figure 1e are adequate to demonstrate this decay to background levels.

The leading velocity kymograph in Figure 1c is very noisy. Does it show anything beyond noise around an average value and was there any effort to calculate the velocity over varying time intervals to smooth it out and possibly reveal a pattern?

As alluded to in our answer to the reviewer’s first comment, the kymographs are not central to any conclusions in the paper, but rather serve as a visual aid for readers to obtain a qualitative sense of the data. We did not investigate the velocity kymograph further, because, as mentioned in the paper (lines 91-92), a noisy velocity kymograph is consistent with the very simple expectation that leading edge shape fluctuations are driven by stochastic fluctuations in actin network growth. Further, as kymographs are ultimately a qualitative method, we chose to analyze the fluctuations using a superior quantitative method: the Fourier autocorrelation analysis that forms the basis for the rest of the manuscript.

2. A model with fluctuating membrane pushed by a dendritic network has been developed by Schaus et al. (Ref. 45) with a follow up paper in Biophys J. A comparison/discussion I feel is necessary, even if the similarities end up superficial. These authors also varied the branching angle.

While the two papers referenced by the reviewer included a model similar to the current work (in that they incorporated stochastic filament polymerization near a flexible membrane), these papers were mainly focused on self-organization of the actin network, largely ignoring the membrane shape. For example, Schaus et al. (2007) designed their model to study the self-organization of actin filament orientation, and Schaus and Borisy (2008) sought to understand limitations on the optimal “performance” and load-sharing of filaments from an engineering perspective. Importantly, these papers simulated only small segments of a leading edge that were < 2 µm – and thus barely touched on questions of leading edge stability and shape fluctuations. Therefore there are no results that could be compared to the current work, and any comparative discussion would potentially mislead and/or confuse the reader. However, we now acknowledge the conceptual similarity of our approach to the referenced publications in the introduction of our model (lines 134-135). Of course, we also reference this work where it is most relevant to our results: in the context of previous literature discussing self-organization of actin filament orientation (lines 246-248, 253-255) – as was recognized by the reviewer – and now with added emphasis on line 255.

3. Intrinsic stability of branched actin networks seems to be a main message of the paper and it's discussed in Figure 4. But why would one expect instability and why was it a surprise that the simple model would lead to stable fluctuations (page 9, beginning of last paragraph)? From Figure 4g, this it seems the authors may have in mind a density/curvature instability, perhaps with a characteristic wavelength (which is yet what appears to be seen in Figure 1b as I mention on comment 1).

We thank the reviewer for raising this point, which prompted us to make several changes to the Introduction that clarify why our result is surprising. For our complete response, please see our reply to the first comment by Reviewer #2, who had a similar question.

Also, given the emphasis on spectral analysis in Figure 1-3, in Figure 4 I was expecting to see how membrane tension and bending influence the amplitude and decay constants of the modes (but this is not shown). Do membrane tension and bending energy play any role in the fluctuations, and at what values? I would anticipate the spectral analysis is most relevant in Figure 4, to show the wavelength-dependence of the stability mechanism in Figure 4j (and possibly its relationship to the spectrum of Figure 2).

We are glad the reviewer takes interest in our spectral analysis. Indeed, we also believe a dissection of the exact shape of the fluctuation amplitude and decay rate curves, as well as their dependence on model parameters such as membrane tension, would be an exciting area for future study. However, the major findings of this work are the emergent stability of lamellipodial actin and the optimal angle of the Arp2/3-complex – neither of which depend on the details of the spectral analysis. Rather, we use the spectral analysis to quantitatively describe the experimentally observed fluctuations and to validate our model by comparison to the experimental data. While further interrogation of the spectra would be interesting, it is not required for any of our main conclusions and is thus beyond the scope of the work.

With respect to the stability mechanism suggested in Figure 4j, it is not trivial to intuit the associated fluctuation spectra from first principles, and therefore any comparison to the results shown in Figure 2 would be speculative. One could develop an analytical model reflecting our proposed mechanism, which could then predict the wavelength-dependence of the proposed stability mechanism and be compared to our simulations. However, we believe our identification of this novel mechanism, on its own, is already a major advance in the field. The additional development and validation of an associated analytical model would be a substantial undertaking and is left for future studies.

4. (Related to the previous question) Is there a qualitative or quantitative interpretation of the behavior of the amplitude of fluctuations versus wavelength in Figure 2k? For example, does this graph show a crossover between two (power-law?) regimes from high amplitude large wavelengths confined by the cell/simulation size to suppressed fluctuations at short wavelengths as a result of the stability mechanism in Figure 4j? I am partly asking this trying to understand the long wavelength behavior as function of the leading edge length in Figure S5 where I would have expected the longest wavelength fluctuations to increase with system size.

As discussed in our answer to the reviewer’s previous comment, we believe an in-depth investigation into the exact shape of the fluctuation spectrum curves is beyond the scope of the current work and is not required to inform the major conclusions of our study. The observed curves could possibly be determined by any number of complex interactions between the relevant variables (polymerization velocity, actin density, actin filament orientation, membrane tension/bending rigidity, etc.) – and so even a qualitative interpretation would be speculative at best. Therefore we did not include any speculation in the text of the manuscript.

With regards to the reviewer’s specific reference to Figure S5 (now Figure S6), the fluctuation amplitudes for each wavelength are not, and should not be, constrained by the length of the leading edge. If we simulate a longer leading edge, we can resolve longer wavelength fluctuations, but the amplitude of any particular mode should not be affected by the size of the system (length of the leading edge). The size of the system is a simulation parameter, not a biological parameter, so it should not affect the result (i.e., a simulated small patch of membrane should behave identically to an equivalently sized portion of a larger simulated patch of membrane). Figure S5 (now Figure S6) is therefore a control demonstrating that our results correctly do not depend on the leading edge length simulated. To clarify these points for the reader, we have added an explanation for each of the control simulations to the Methods (lines 477-482).

5. Various modeling works have shown that the filament angle distribution at steady state have different preferred orientation, depending on both the branching angle as well as the ratio of protrusion to polymerization rate (which determine if a polymerizing branch can catch up with the leading edge). I assume that as the branching angle is varied in Figure 5, the system may change preferred steady state orientations. As I understand, the analysis in Figure 5 showing that the genetically-encoded Arp2/3 angle is optimal was performed for fixed polymerization rate. Wouldn't the optimal value change depending on the polymerization rate?

We are excited that the reviewer is curious about the mechanisms by which the Arp2/3-mediated branching angle optimally smooths fluctuations – a topic that we believe will be of considerable interest for follow-up studies based on this work.

The reviewer brings up an interesting question about the role of the polymerization rate in leading edge fluctuations. However, we do not believe there is a trivial relationship between the polymerization rate and the optimal branching angle. While the relationship between branching angle, the preferred filament orientation, and polymerization velocity is well understood, how or whether any of these variables might affect the leading edge fluctuation amplitude remains entirely unclear*.* If we cannot a priori predict how a change in the polymerization velocity should affect the fluctuation amplitude for a *given* branching angle, then we certainly cannot predict how a change in the polymerization velocity would affect the minimum amplitude across all possible branching angles. Given these complexities, any investigation or discussion of the effect of the polymerization rate – however interesting – would be beyond the scope of the current work.

In any case, the polymerization rate used in our simulations was specifically chosen to be within a biologically-relevant regime and well-grounded in experimental data. The rate was taken directly from published experimental measurements (Table S1), and produces a simulated leading edge velocity nearly identical to that of our experimentally measured cell velocities (Figure 1c, Figure 2g). Therefore the 70-80° branching angle is indeed optimal for the experimentally-measured cell velocities and actin polymerization rates – which should be most relevant for the question we were most interested in, regarding the evolutionary conservation of the branching angle in cells. We emphasize that this is the first mechanistic evidence *of any kind* regarding the optimality of the evolutionarily conserved angle, and thus represents a major advance that stands on its own. Any explorations into the roles of the polymerization rate on the optimal branching angle would be outside the scope of this work.

Minor points6. In Table S1, the actin filament persistence length seems very small, 1 micron. This is an important parameter for this Brownian ratchet model. Is this a typo?

We thank the reviewer for their keen eye – however, our use of 1µm as the persistence length is not a typo. We use the same value for the persistence length as Mogilner and Oster (1996), and as measured by analysis of thermal undulations of actin filaments in Käs et al. (1996) (Käs et al., 1996; Mogilner and Oster, 1996). While we acknowledge that a persistence length of ~10-20 µm is a more commonly cited range, all of the known experimentally measured values have been acquired using in vitro systems, rather than in cells. Given recent evidence that microtubules exhibit a persistence length ~100 times smaller in cells than those observed in vitro (Pallavicini et al., 2014), we believe it is not unreasonable to assume a value on the lower end of in vitro experimental measurements for Factin.

7. An outline or plan of the Appendix material would be helpful to understand the purpose of various calculations involved as one starts reading it. The math can also greatly be reduced, for example I don't think showing the details of a simple integration such as Equation (55) is needed, this takes away from focusing on more important equations such as 56.

We thank the reviewer for their suggestion to add an outline of the Appendix material. An introductory paragraph has been added at the beginning of each Appendix in the Supplementary Text.

We respectfully disagree that the detailed derivations should be removed. As this manuscript was written for the broad readership of the Journal, we believe that not all readers will find Supp. Equation 55 to be simple – particularly those without a strong background in math and physics, many of whom will not be familiar with the error function, for example. We find it is far easier for those having familiarity with the material to skip over seemingly trivial details, than for someone with no familiarity to fill in the gaps themselves.

8. A simulation movie at shorter time intervals than the one provided might be more helpful: the current one shows snapshots that differ significantly from frame to frame.

The purpose of this video was to display the simulations at a similar timescale to that shown for the experimental data – in particular, to demonstrate the long timescale stability of the simulated leading edges. Therefore we show the full simulated time (100s) at 0.5 s resolution. Displaying at faster time resolution would quickly make the file size prohibitively large.

9. It may be helpful to list somewhere the values of the wavelength of the modes in Figure 1e,f and elsewhere. At first reading I understood these curves to be the first few modes rather than a sampling of different modes.

We thank the reviewer for prompting us to clarify the wavemodes shown in Figure 1e-f. The spatial frequencies are equally spaced between 0.22-0.62 µm^-1^ in 0.06 µm^-1^ intervals, corresponding to wavelengths between 1.6-4.5 µm. This information has been added to the legend for Figure 1.

10. Figure 5: is there a reason to choose branching angles of 17.5, 35 and 70 deg in 5c while 17.5, 70 and 140 degrees in 5d?

A different set of angles were displayed in Figure 5c, as compared to Figure 5d, as the two graphs are making a different set of comparisons. Figure 5c shows not only how the dominant angle is always one half of the branching angle, θ_br_, but also show additional resonant peaks at θ_br_/2 + θ for branches growing off of filaments lying at the dominant angle. Given the axes limits of 0-90° (filaments pointing towards the membrane), these resonant peaks can only be seen for branch angles less than 60°. In contrast, Figure 5d-e shows that a 70° branching angle optimally minimizes actin density variability, for which angles both above and below 70° must be displayed for comparison.

11) In case this helps, I noticed the κ_eff_ above Equation (49) has the numerical prefactor in Dickinson Biophys J 87:2838 (2004) that they mention is different to Mogilner and Oster, though I am not sure if the calculation here is the same.

We thank the reviewer again for their keen eye in drawing our attention to the fact that Dickinson et al. also arrived at this prefactor. As discussed in Dickinson et al., Mogilner and Oster arrive at a different prefactor because they approximate filament bending to occur only along an arc of constant curvature, rather than treating the full spectrum of bending fluctuations – as performed in this work and in Dickinson et al. We have added a discussion to this extent to the Supplementary Text associated with Equation (49).

References:

Crow, J. F., and Kimura, M. (1970). An introduction to population genetics theory*. An introduction to population genetics theory.* New York, Evanston and London: Harper and Row, Publishers.

Diz-Muñoz, A., Fletcher, D. a., and Weiner, O. D. (2013). Use the force: Membrane tension as an organizer of cell shape and motility. *Trends in Cell Biology*, *23*(2), 47–53. https://doi.org/10.1016/j.tcb.2012.09.006

Imhof, M., and Schlotterer, C. (2001). Fitness effects of advantageous mutations in evolving *Escherichia coli* populations. *Proceedings of the National Academy of Sciences*, *98*(3), 1113–1117. https://doi.org/10.1073/pnas.98.3.1113

Käs, J., Strey, H., Tang, J. X., Finger, D., Ezzell, R., Sackmann, E., and Janmey, P. A. (1996). F-actin, a model polymer for semiflexible chains in dilute, semidilute, and liquid crystalline solutions. *Biophysical Journal*, *70*(2 I), 609–625. https://doi.org/10.1016/S0006-3495(96)79630-3

Liu, X., Oh, S., Peshkin, L., and Kirschner, M. W. (2020). Computationally enhanced quantitative phase microscopy reveals autonomous oscillations in mammalian cell growth. *Proceedings of the National Academy of Sciences of the United States of America*, *117*(44), 27388–27399. https://doi.org/10.1073/pnas.2002152117

Maly, I. V., and Borisy, G. G. (2001). Self-organization of a propulsive actin network as an evolutionary process. *Proceedings of the National Academy of Sciences*, *98*(20), 11324–11329. https://doi.org/10.1073/pnas.181338798

Mogilner, A., and Oster, G. (1996). Cell motility driven by actin polymerization. *Biophysical Journal*, *71*(6), 3030–3045. https://doi.org/10.1016/S0006-3495(96)79496-1

Mullins, R. D., Bieling, P., and Fletcher, D. A. (2018). From solution to surface to filament: actin flux into branched networks. *Biophysical Reviews*, *10*(6), 1537–1551. https://doi.org/10.1007/s12551018-0469-5

Narayanan, J., Xiong, J. Y., and Liu, X. Y. (2006). Determination of agarose gel pore size: Absorbance measurements vis a vis other techniques. *Journal of Physics: Conference Series*, *28*(1), 83–86. https://doi.org/10.1088/1742-6596/28/1/017

Pallavicini, C., Levi, V., Wetzler, D. E., Angiolini, J. F., Benseñor, L., Despósito, M. A., and Bruno, L. (2014). Lateral motion and bending of microtubules studied with a new single-filament tracking routine in living cells. *Biophysical Journal*, *106*(12), 2625–2635. https://doi.org/10.1016/j.bpj.2014.04.046

Parekh, S. H., Chaudhuri, O., Theriot, J. A., and Fletcher, D. A. (2005). Loading history determines the velocity of actin-network growth. *Nature Cell Biology*, *7*(12), 1219–1223. https://doi.org/10.1038/ncb1336

Pipathsouk, A., Brunetti, R., Town, J., Breuer, A., Pellett, P., Marchuk, K., … Weiner, O. (2019). WAVE complex self-organization templates lamellipodial formation. *BioRxiv*. https://doi.org/10.1101/836585

Prass, M., Jacobson, K., Mogilner, A., and Radmacher, M. (2006). Direct measurement of the lamellipodial protrusive force in a migrating cell. *Journal of Cell Biology*, *174*(6), 767–772. https://doi.org/10.1083/jcb.200601159

Risca, V. I., Wang, E. B., Chaudhuri, O., Chia, J. J., Geissler, P. L., and Fletcher, D. a. (2012). Actin filament curvature biases branching direction. *Proceedings of the National Academy of Sciences*, *109*(8), 2913–2918. https://doi.org/10.1073/pnas.1114292109

Schaus, T. E., and Borisy, G. G. (2008). Performance of a population of independent filaments in lamellipodial protrusion. *Biophysical Journal*, *95*(3), 1393–1411. https://doi.org/10.1529/biophysj.107.125005

Schaus, T. E., Taylor, E. W., and Borisy, G. G. (2007). Self-organization of actin filament orientation in the dendritic- nucleation/array-treadmilling model. *Proceedings of the National Academy of Sciences of the United States of America*, *104*(17), 7086–7091. https://doi.org/10.1073/pnas.0701943104

Sens, P., and Plastino, J. (2015). Membrane tension and cytoskeleton organization in cell motility. *Journal of Physics: Condensed Matter*, *27*(27), 273103. https://doi.org/10.1088/09538984/27/27/273103

Svitkina, T. M., and Borisy, G. G. (1999). Arp2/3 complex and actin depolymerizing factor/cofilin in dendritic organization and treadmilling of actin filament array in lamellipodia. *Journal of Cell Biology*, *145*(5), 1009–1026. https://doi.org/10.1083/jcb.145.5.1009

Zegers, M. M., and Friedl, P. (2015). Translating Membrane Tension into Cytoskeletal Action by FBP17. *Developmental Cell*, *33*(6), 628–630. https://doi.org/10.1016/j.devcel.2015.06.006